# TMEM16K is an interorganelle regulator of endosomal sorting

Maja Petkovic [1✉], Juan Oses-Prieto [2], Alma Burlingame[2], Lily Yeh Jan[1,3] & Yuh Nung Jan [1,3✉]

Communication between organelles is essential for their cellular homeostasis. Neurodegeneration reflects the declining ability of neurons to maintain cellular homeostasis over a lifetime, where the endolysosomal pathway plays a prominent role by regulating protein and lipid sorting and degradation. Here we report that TMEM16K, an endoplasmic reticulum lipid scramblase causative for spinocerebellar ataxia (SCAR10), is an interorganelle regulator of the endolysosomal pathway. We identify endosomal transport as a major functional cluster of TMEM16K in proximity biotinylation proteomics analyses. TMEM16K forms contact sites with endosomes, reconstituting split-GFP with the small GTPase RAB7. Our study further implicates TMEM16K lipid scrambling activity in endosomal sorting at these sites. Loss of TMEM16K function led to impaired endosomal retrograde transport and neuromuscular function, one of the symptoms of SCAR10. Thus, TMEM16K-containing ER-endosome contact sites represent clinically relevant platforms for regulating endosomal sorting.

[1] Departments of Physiology, Biochemistry and Biophysics, University of California at San Francisco, San Francisco, CA 94158, USA. [2] Department of Pharmaceutical Chemistry, University of California San Francisco, San Francisco, CA 94158, USA. [3] Howard Hughes Medical Institute, University of California, San Francisco, CA, USA. ✉email: Maja.Petkovic@ucsf.edu; YuhNung.Jan@ucsf.edu

Cellular organelles do not act as discrete autonomous units, but rather as interconnected hubs that engage in extensive communication to coordinate their function and maintain cell homeostasis over a cell's lifetime. An emerging theme is that such coordination can be mediated via membrane contact sites (MCS) between distinct organelles[1–3]. MCS are specialized microdomains in which organelles are held by tethers in close proximity to one another without fusing. Such interorganelle tethers are formed by a variety of complexes composed of membrane integral proteins, peripherally-associated proteins as well as aided by specific lipids like phosphatidylinositols[4,5]. Our knowledge of cellular processes taking place through interorganelle communication at contact sites is continuously expanding. MCS were shown to be sites of phospholipid biosynthesis[6] and lipid transfer[7] between the two apposing membranes regulating lipid metabolism[3,8]. MCS are the sites of calcium transfer[9] between organelles regulating calcium homeostasis[10]. They are required for organelle biogenesis[11–14], organelle dynamics[15], and signaling[16,17]. Several human orthologues of these MCS proteins have been linked to a broad range of age-related pathologies[18,19], putting these evolutionarily conserved cellular pathways into spotlight as central to cellular physiology. However, the molecular identity and physiological significance of interorganelle communication is still emerging.

The TMEM16 family of proteins is evolutionarily conserved with family members found in all eukaryotes[20], from amoeobozoa[21] and fungi[22,23] to mammals. Fungi and plants often have only one or two TMEM16 family members, while *Caenorhabditis elegans* has two and *Drosophila* has five TMEM16 family members[20]. In mammals, the TMEM16 family comprises ten members, which act as modulators of diverse cellular functions throughout the body and are linked to a variety of genetic disorders, highlighting their pathophysiological importance[24,25]. The TMEM16 family includes the long sought after calcium activated chloride channels[26–28], and many family members across phylogeny are calcium-activated lipid scramblases[21–23,29] mediating the translocation of phospholipids between the leaflets of the membrane bilayer down their concentration gradients.

Interestingly, the single TMEM16 family member in yeast, Ist2p, was one of the first reported MCS tethers shown to play a vital role in lipid homeostasis at contact sites between the endoplasmic reticulum (ER) and plasma membrane[30–32]. Given the biophysical properties and cellular functions of its mammalian homologs, where they act at the convergence of numerous cellular pathways, an exciting hypothesis for exploration concerns the possibility that they similarly participate in interorganelle communication. Yet, outside of the yeast studies, TMEM16 family members have been extensively investigated thus far for roles other than those at membrane contact sites. To evaluate their potential role in interorganelle communication we focus on the lipid scramblase TMEM16K[33], the least divergent member of the mammalian family[25] (Supplementary Fig. 1a) responsible for an autosomal recessive form of progressive neurodegenerative disease, spinocerebellar ataxia (SCAR10)[34–36].

Here, we find that TMEM16K knockout mice display defects in neuromuscular function and motor behaviors, corresponding to ataxic phenotypes observed in human patients. Loss of TMEM16K leads to impaired endosomal retrograde trafficking and dysfunction in the endolysosomal pathway. We find endoplasmic reticulum-localized TMEM16K acts at ER-endosome contact sites where it interacts with the endosomal protein Rab7. Reintroduction of wild type TMEM16K, but not human disease variants rescues the observed cellular defect. We conclude TMEM16K is an interorganelle regulator of endosomal sorting.

## Results

**TMEM16K knockout mice display progressive impairment in neuromuscular function.** We generated mouse models with either ubiquitous or neuron specific loss of TMEM16K (Fig. 1a) to evaluate if the pathology is conserved between mouse and human. As impairment of neuromuscular function is a classical symptom of ataxia, we analyzed neuromuscular junctions (NMJ)[37] in TMEM16K knockout mice at 6 and 24 months of age. Using bungarotoxin staining as a marker for NMJ, we found a progressive reduction in the size of the NMJ (Fig. 1b, c). Moreover, knockout mice displayed increasing hindlimb clasping, a behavioral phenotype marking disease progression in a number of mouse models of neurodegeneration[38,39] (Fig. 1d, Supplementary Movie 1). As TMEM16K is broadly expressed[40,41] (Supplementary Fig. 1b), we analyzed neuron specific TMEM16K knockout mice and wild type littermates at 24 months of age to evaluate whether loss of TMEM16K in neurons is sufficient to cause the observed phenotypes. These animals lacking neuronal TMEM16K displayed increased hindlimb clasping, as well as an impaired ability to complete a ledge-walking test (Fig. 1e). Together, these results demonstrate a phenotypic linkage between loss of TMEM16K and impaired neuromuscular function that is conserved between mice and human.

**The TMEM16K protein interaction network.** TMEM16K is localized to the endoplasmic reticulum[33], a localization shared with its yeast[42] and *Drosophila*[43] homologs. We also found that TMEM16K localizes to the ER as evident by its colocalization with several established endoplasmic reticulum markers (Fig. 1f, g, Supplementary Movie 2). To find the potential cellular functions of TMEM16K in an unbiased manner, we set out to identify the TMEM16K protein interaction network using proximity-dependent biotinylation[44–46] (BioID). This approach uncovers direct and indirect interactions within a 10-nm range of the promiscuous biotinylation enzyme tagged to the protein of interest (Fig. 2a). We tagged TMEM16K with biotin ligase and confirmed retention of both endoplasmic reticulum localization (Supplementary Fig. 2a, b) and biotinylation activity (Supplementary Fig. 2c), permitting the identification of TMEM16K-proximal proteins by mass spectrometry of affinity-purified biotinylated proteins from transfected cells (Fig. 2a). We obtained a list of potential TMEM16K interactors (Fig. 2b, Supplementary Data 1) and, instead of hand-picking a few candidates, we visualized this list as a protein–protein interaction network to identify the most biologically interconnected clusters of proteins, which could infer TMEM16K function (Fig. 2b). First, we calculated protein–protein interaction enrichment to determine if the obtained candidate list has more or less interactions among themselves, as compared with a random set of proteins of similar size. Protein–protein interaction enrichment $p$ value of the TMEM16K network is $p < 1.0E{-}16$, suggesting biological connection of proteins that interact with TMEM16K. Next, we performed functional enrichment analysis, and overlaid the major functional categories on our candidates, suggesting the presence of functional clusters in our candidate list. Hence, we performed clustering analysis to bioinformatically identify such clusters, defined as highly interconnected nodes, and generated a simplified network of TMEM16K major clusters overlaid with functional enrichment categories (Fig. 2c). As expected, when evaluating a protein over its lifetime, we found clusters involved in protein processing and degradation. Consistent with the function of its *Drosophila* homolog[43], we also identified a cluster of proteins involved in nuclear organization. Unexpectedly, this analysis revealed that endosomal transport, in particular

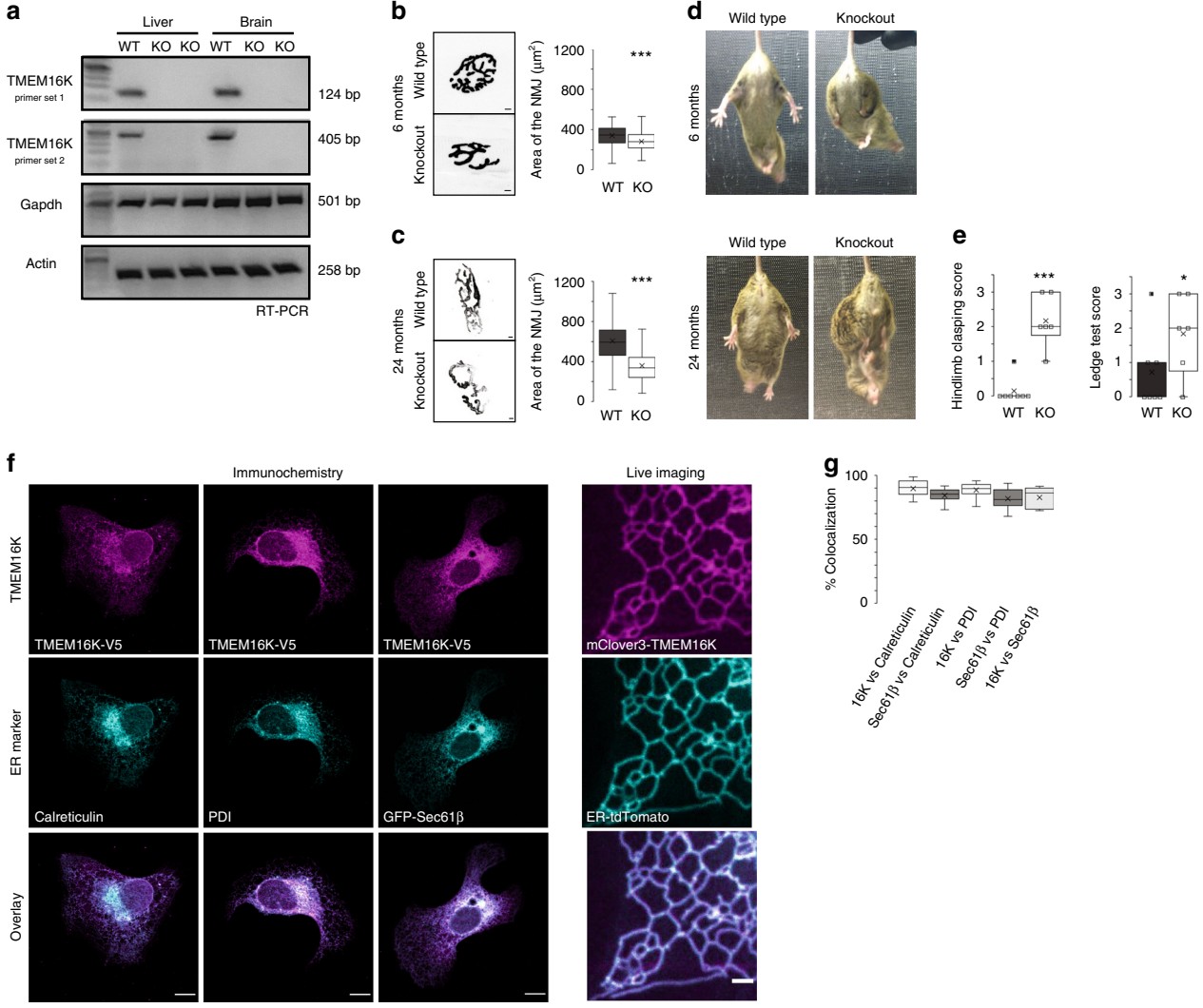

**Fig. 1 TMEM16K knockout mice. a** RT-PCR from liver and brain tissues obtained from the wild type and TMEM16K full knockout mice. Two different sets of primers amplifying TMEM16K were used, and Gadph and β-actin were amplified as controls. **b** Representative images and quantification of neuromuscular junction (NMJ) at 6 months of age from wild type ($n = 133$ NMJ, three animals) and TMEM16K full KO ($n = 132$ NMJ, four animals) littermates visualized with the fluorescently labeled α-Bungarotoxin. Scale bar 5 μm, Single factor ANOVA, $p$ value = 2.10E−06*** **c** Representative images and quantification of neuromuscular junction at 24 months of age from WT ($n = 137$ NMJ, five animals) and TMEM16K full KO ($n = 126$ NMJ, four animals) littermates. Scale bar 5 μm, Single factor ANOVA, $p$ value = 1.31E−22*** **d** Representative images of hindlimb clasping, of WT and TMEM16K full KO littermates at 6 and 24 months of age. **e** Quantification of hindlimb clasping ($p$ value = 0.00003***) and ledge walking ($p$ value = 0.05*) at 24 months of age of neuron specific TMEM16K KO ($n = 6$ animals) and their WT ($n = 7$ animals) mice. One-tailed $t$-test. See Supplementary Movie 1. **f** Columns 1–3: Immunocytochemistry of U-2OS cells transfected with TMEM16K tagged with V5 epitope and stained for ER-markers Calreticulin, Protein Disulfide-Isomerase (PDI) and Sec61β, respectively. Scale bar 10 μm. **g** Column 4: Snapshot from live imagining of COS-7 cells expressing with TMEM16K tagged with mClover3 and ER-tdTomato. Scale bar 2 μm. See Supplementary Movie 2. **i** Quantification of the colocalization of TMEM16K and ER-markers Calreticulin, PDI and Sec61β using Mander's overlap coefficient, as well as quantification of the colocalization of Sec61β with Calreticulin and PDI measured in the same manner. Sec61β colocalization with other ER markers is included to provide a meaningful context for the colocalization analysis with TMEM16K, given that Sec61β is a pore forming component of the translocon complex localized exclusively to the ER (three biological replicates, $n = 38$ cells for TMEM16K vs PDI, 44 for TMEM16K vs Calreticulin, 20 for TMEM16K vs Sec61β, 42 for Sec61β vs Calreticulin, 38 for Sec61β vs PDI). In the box and whiskers plot, the box includes the first quartile and the third quartile, with the central line representing the median. Whiskers represent the minimum and maximum values of data. X inside the box represents the mean of data. Source data are provided as a Source Data file.

endosomal retrograde trafficking, is a major cluster in the TMEM16K network (Fig. 2c).

**TMEM16K is required for endosomal retrograde transport.** Dysfunctions of endosomal transport are tightly associated with neurodegenerative diseases[47,48]. As the TMEM16K interactome pointed to endosomal retrograde transport, we investigated whether TMEM16K is required for proper trafficking of the cation-independent mannose-6-phosphate receptor (CI-MPR), which is the best-studied retrograde-transport cargo in mammals[49,50]. In wild type primary mouse embryonic fibroblasts (MEF), when an antibody that recognizes the extracellularly exposed CI-MPR is pulse chased from the plasma membrane, it gets internalized in the endosomes and subsequently transported through the endosomal retrograde pathway to the perinuclear region corresponding to the trans-Golgi network (TGN) within 60 min[1,2] (Fig. 3a). However, in MEF from TMEM16K knockout

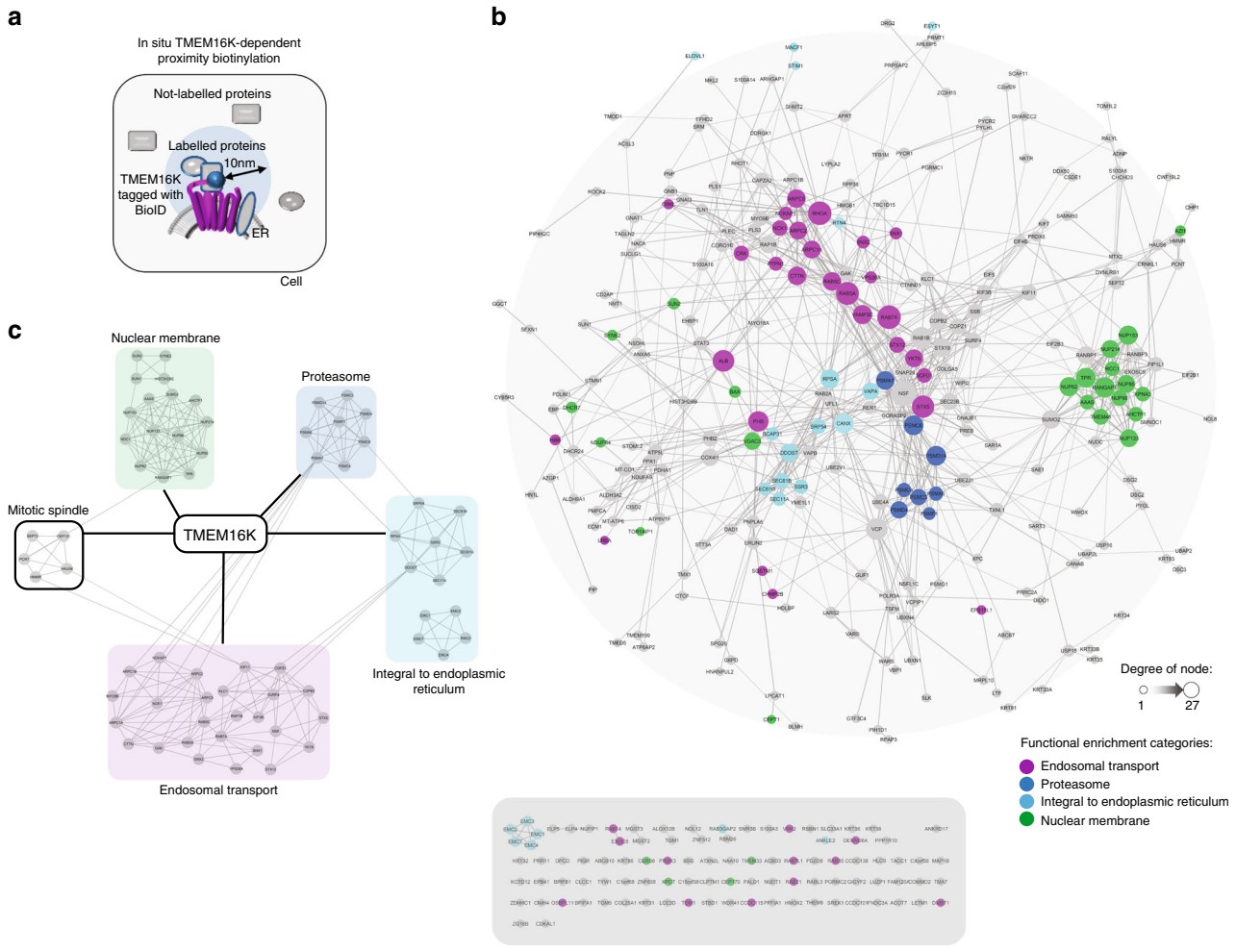

**Fig. 2 Proteomic mapping of TMEM16K via in situ BioID-catalyzed biotin labeling finds endosomal transport as a major functional cluster. a** Scheme of proteomic mapping of protein complexes surrounding TMEM16K in the radius of 10 nm via in situ proximity biotinylation. **b** TMEM16K proteome candidate list is visualized as a protein-protein interaction network using the String protein interaction public database in Cytoscape. Candidates without known protein-protein interactions in the String database are represented on the bottom in the gray panel. TMEM16K is omitted from this representation for simplicity. Functional enrichment based on the GO terms was calculated using the String app in the Cytoscape and the major identified categories of functional enrichment were overlayed on our candidates with color-code. Purple: Endosomal transport (False Discovery Rate (FDR) $p$ value = 2.49E−4), endosome to Golgi retrograde trafficking (FDR $p$ value = 0.0096); Cyan: ER membrane protein complex (FDR $p$ value=1.5E-5), protein localization to endoplasmic reticulum (FDR $p$ value = 0.0026); Green: nuclear membrane (FDR $p$ value = 1.28E−6), nuclear pore (FDR $p$ value = 3.38E−9); Blue: proteasome (FDR $p$ value = 5.74E−5). **c** Bioinformatic analysis of the TMEM16K candidates list with MCODE cluster app in Cytoscape identified major clusters in our dataset, which generated simplified network of TMEM16K proteomics data. Color coding of functional enrichment analysis was overlaid on the bioinformatically identified clusters. TMEM16K candidate list is provided in Supplementary Data 1.

mice, the internalized antibody against CI-MPR remained dispersed peripherally during the 60 min period of pulse chase, consistent with a defect in endosome to trans-Golgi retrograde trafficking (Fig. 3a). Reintroduction of TMEM16K rescued this CI-MPR retrograde trafficking defect of mutant MEF (Fig. 3a). As this pathway is also co-opted by a subgroup of pathogens during their entry into cells, we used cholera toxin B (CTxB)[51] to further corroborate our finding. Indeed, following CTxB internalization, TMEM16K knockout cells displayed reduced CTxB colocalization with the Golgi marker GM130 after 60 min of pulse chase, which can be rescued by reintroduction of TMEM16K (Fig. 3b), confirming that TMEM16K is required for proper endosome to trans-Golgi retrograde trafficking. However, no change was observed in the localization of Golgi complex proteins, or in the morphology of the Golgi complex (Fig. 3b; Supplementary Fig. 3a–c), suggesting that loss of TMEM16K function does not affect the Golgi complex. Altogether, these data demonstrate that

depletion of the ER-resident protein TMEM16K perturbs endosomal retrograde trafficking, a defect similar to that observed upon depletion of known cargo-sorting components[50,52,53].

**Loss of TMEM16K causes defects in endosomal sorting and acidification.** As observed defects in endosomal retrograde transport could be due to perturbations in multiple parts of the pathway, we set out to determine at which point in the pathway is TMEM16K required. To ensure that the observed defect in endosomal retrograde transport is not due to impaired anterograde secretory pathway, we took advantage of an approach that allows synchronization of protein transport through the biosynthetic pathway[54,55]. Using this RUSH system we tracked the biosynthetic transport of three transmembrane proteins with different steady state distributions: the glycosylphosphatidylinositol anchor (GPI; transported to plasma membrane), the transferrin

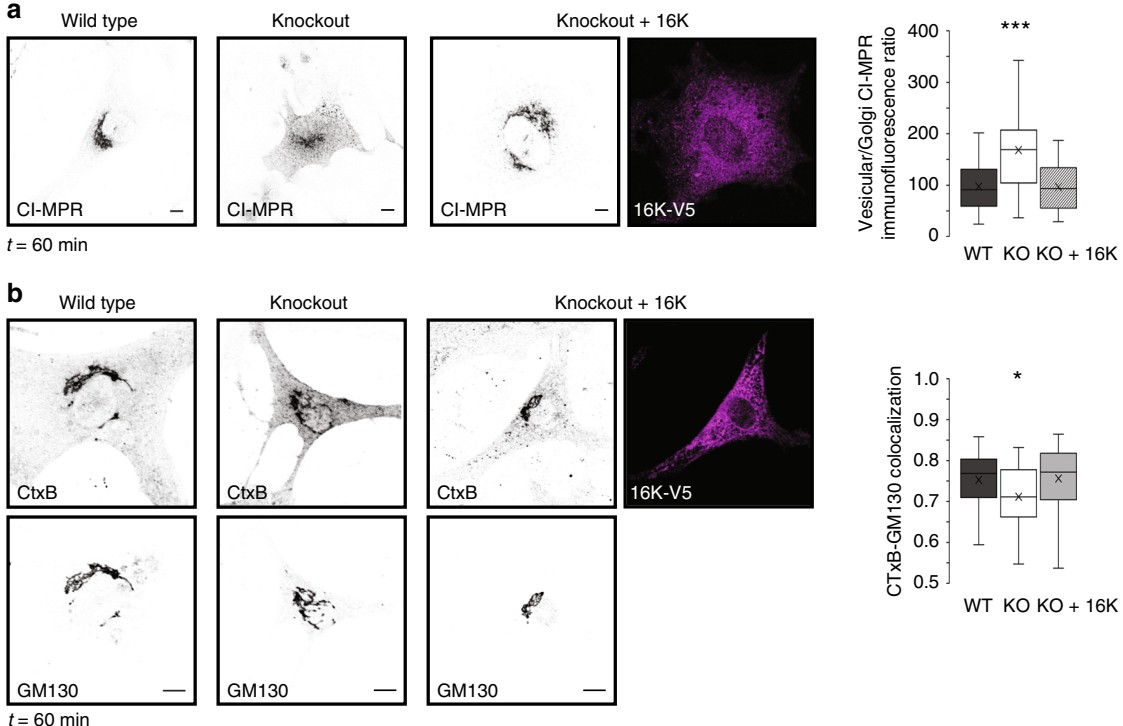

**Fig. 3 TMEM16K requirement for endosomal retrograde transport. a** Left, Immunofluorescence of pulse chased antibody detecting CI-MPR internalized from the plasma membrane at 60 min time point in the WT, TMEM16K KO cell and TMEM16K KO cell with reintroduced TMEM16K. Scale bar 10 μm. Right, Ratio of measured intensity between vesicular region of the cell and the region encompassing Golgi (10 × 10 μm$^2$). Single factor ANOVA, $p$ value = 4.65E−37, post-test Bonferroni-corrected two sided $t$-test with pairwise comparison with WT (three biological replicates, $n$ = 168 WT, 181 TMEM16K KO, $p$ value = 1.37E−26***, and 134 KO + 16K cells). **b** Left, Immunofluorescence of Golgi marker GM130 and internalized conjugated cholera toxin B (CtxB) from the plasma membrane at 60 min time point in the WT, TMEM16K KO cell and TMEM16K KO cell with reintroduced TMEM16K. Scale bar 10 μm. **b** Right, Quantification of the Pearson's correlation coefficient measuring colocalization of GM130 and CTxB. Single factor ANOVA, $p$ value = 0.0018 with post-test Bonferroni-corrected two sided $t$-test with pairwise comparison with WT (three biological replicates, $n$ = 40 WT, 57 TMEM16K KO, $p$ value = 0.0051*, and 56 KO + 16K cells). In the box and whiskers plot, the box includes the first quartile and the third quartile, with the central line representing the median. Whiskers represent the minimum and maximum values of data. X inside the box represents the mean of data. Source data are provided as a Source Data file.

receptor (TfR; transported to plasma membrane, early endosomes, and recycling endosomes), and the cation-dependent mannose-6-phosphate receptor (CD-MPR; transported from TGN directly to early/late endosomes). We found no difference in the transport through the biosynthetic pathway between TMEM16K wild type and knockout cells (Fig. 4a–c), showing that the anterograde secretory pathway is unaffected.

To look for evidence for potential defects upstream of the endosomal retrograde sorting in the endolysosomal pathway, we first evaluated whether endocytosis is affected by examining the internalization of fluorescently labeled transferrin. We found no differences between TMEM16K wild type and knockout cells (Fig. 4d). Next, we performed EGF pulse-chase experiments to further evaluate maturation from early to late endosomes. We found no difference between the wild type and knockout cells in the colocalization of EGF with the late endosomal marker Rab7 at the 10, 15, and 40 min time points (Fig. 4e), indicating that the mutant phenotype arose from a defect at or after the Rab7 stage of endolysosomal maturation. These results show that the upstream endosomal pathway is unaffected in TMEM16K knockout cells. However, at 60 min a larger fraction of EGF was retained in Rab7 endosomes in TMEM16K knockout cells, compared with wild type cells (Fig. 4e), suggesting defect in endosomal sorting.

To evaluate whether the endolysosomal pathway downstream of endosomal sorting was perturbed, we examined a major

distinguishing feature of endolysosomal maturation, acidification. Using Lysosensor Green DNP-189, a fluorescent pH indicator that partitions into acidic organelles, we found that acidification was impaired in the absence of TMEM16K (Fig. 4f). To confirm that the differences in DNP-189 fluorescence reflected differences in pH within organelles, we utilized the protonophore FCCP to selectively eliminate the pH gradient. Consistent with an inability to form and/or maintain a proton gradient, TMEM16K knockout cells displayed both a slower rate of FCCP-induced proton leak and an impaired ability to stabilize proton loss compared with wild type cells (Fig. 4g). To further test the TMEM16K involvement in the observed defect, we performed rescue experiments with wild type TMEM16K or mutant TMEM16K with substitutions of the conserved calcium binding acidic residues required for protein function (E448Q/D497N/E500Q/E529Q/D533N)[33,56,57]. We expressed wild type or mutant TMEM16K in primary cells from wild type or TMEM16K knockout mice. Transfecting mutant TMEM16K into wild type cells yielded dominant negative effect. Expression of wild type, but not mutant TMEM16K in primary cells lacking TMEM16K rescued the acidification defect, demonstrating that TMEM16K is required for normal maturation of endolysosomal compartments (Fig. 4h). Taken together, our results show that loss of TMEM16K causes a defect in endosomal retrograde sorting, and deficiencies within the later stages of the endolysosomal system.

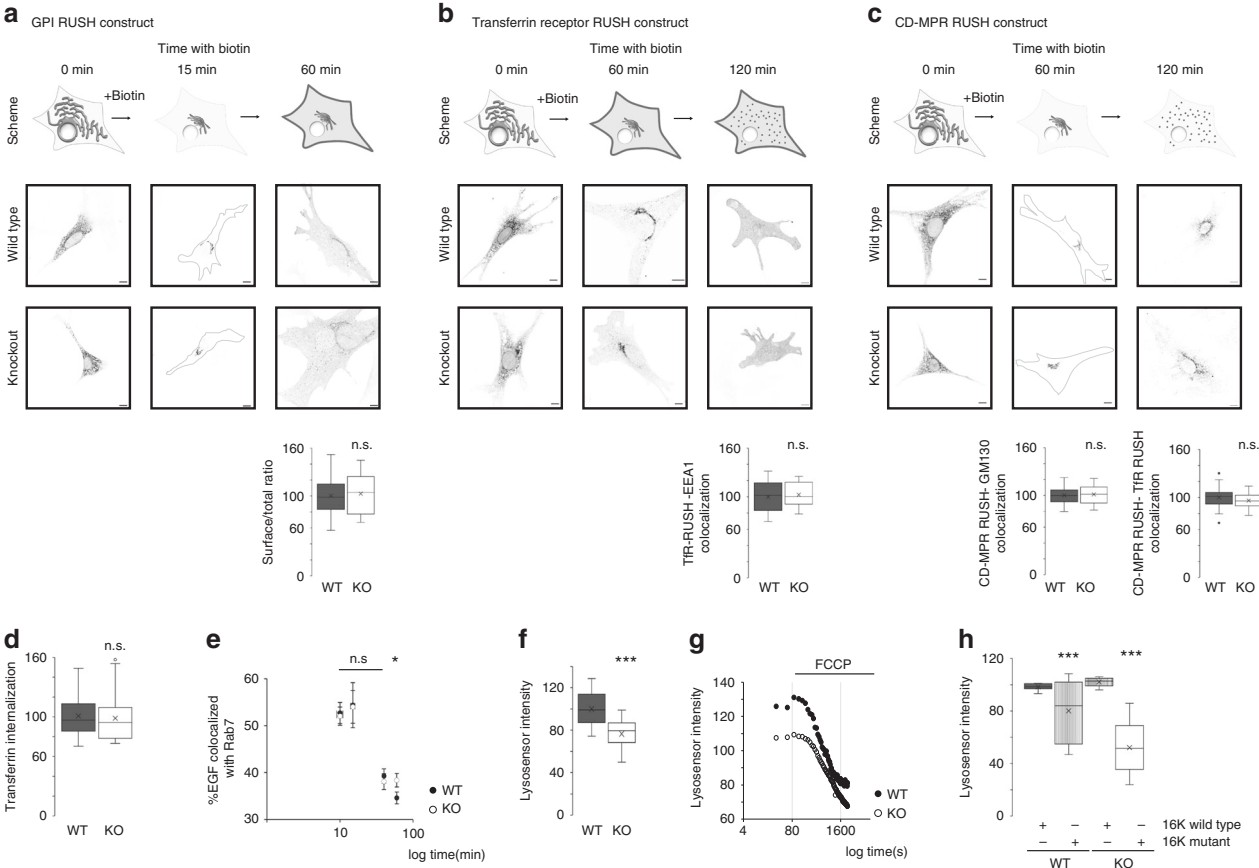

**Fig. 4 Analysis of the endolysosomal pathway in the TMEM16K absence.** RUSH assay 1st row: Scheme of the RUSH construct biosynthetic pathway. 2nd and 3rd rows: Representative images of WT and TMEM16K KO cells at the indicated time points. 4th row: Quantification at indicated time point. **a** RUSH assay with mCherry-GPI. Quantification of surface vs. total at 60 min. Two-tailed $t$-test, $p$ value = 0.53 n.s. (three biological replicates, $n$ = 53 WT and 56 TMEM16K KO cells) **b** RUSH assay with mCherry-TfR. Pearson's correlation coefficient at 120 min. Two-tailed $t$-test, $p$ value = 0.21 n.s. (three biological replicates, $n$ = 129 WT, and 111 TMEM16K KO cells) **c** RUSH assay with GFP-CD-MPR. Pearson's correlation coefficient at 60 min with GM130 (three biological replicates, $n$ = 126 WT, and 127 TMEM16K KO cells, Two-tailed $t$-test, $p$ value = 0.45 n.s.), and Pearson's correlation coefficient at 120 min with mCherry-TfR RUSH (three biological replicates, $n$ = 144 WT and 134 TMEM16K KO cells, Two-tailed $t$-test, $p$ value = 0.17 n.s.) **d** Fluorescence intensity of transferrin at 60 min in the WT ($n$ = 100) and TMEM16K KO ($n$ = 50) cells from three biological replicates, Two-tailed $t$-test, $p$ value = 0.63 n.s. **e** EGF-Alexa647 pulse-chase experiment was quantified for colocalization with endogenous Rab7, Two-tailed $t$-test between WT and KO at each measured time point (three biological replicates, $n$ = 130 WT, and 117 TMEM16K KO cells at 10 min, $p$ value = 0.75 n.s., 91 WT, and 103 KO cells at 15 min, $p$ value = 0.88 n.s., 77 WT and 61 KO cells at 40 min, $p$ value = 0.60 n.s., 89 WT, and 118 KO cells at 60 min, $p$ value = 0.043*). **f** Fluorescence intensity of Lysosensor Green DNP-189 in WT ($n$ = 114) and TMEM16K KO ($n$ = 116) cells. Single factor ANOVA $p$ value = 8E−25*** from three biological replicates.
**g** Representative trace from three independent experiments of protonophore FCCP at a final concentration 2 μM added at 120 s to cells loaded with Lysosensor Green DNP-189. (WT slope is −0.0445, $y$ = −0.0445$x$ + 128.45, $R^2$ = 0.9537; TMEM16K KO slope is −0.0305, $y$ = −0.0305$x$ + 109.5, $R^2$ = 0.9581) **h** Evaluation of wild type and mutant TMEM16K cDNA ability to rescue acidification defect. WT or TMEM16K KO cells were co-transfected with mCherry-CAAX to visualize transfected cells, and TMEM16K wild type cDNA (TMEM16K-FLAG) or TMEM16K mutant cDNA (Ca5MUT-FLAG) and evaluated for acidification with Lysosensor Green D-189. Single factor ANOVA, $p$ value = 1.46E−39 with post-test Bonferroni-corrected two sided $t$-test with pairwise comparison with WT + 16K wild type (three biological replicates, $n$ = 50 WT + 16K wild type; 50 WT + 16K mutant, $p$ value = 1.85E−07***; 40 KO + 16K wild type; 40 KO + 16K mutant cells, $p$ value = 3.27E−19***). In the box and whiskers plot, the box includes the first quartile and the third quartile, with the central line representing the median. Whiskers represent the minimum and maximum values of data. X inside the box represents the mean of data. Source data are provided as a Source Data file.

**TMEM16K forms contacts with Rab7-positive endosomes.** The requirement of TMEM16K for endosomal retrograde trafficking raised the question how this endoplasmic reticulum localized protein affects endosomes. Given that proteomics revealed that TMEM16K is in the proximity of the endosomal compartment for direct or indirect interactions, we considered the possibility that TMEM16K facilitates endosomal sorting through inter-organelle communication at the sites of contact between the ER and endosomes. Membrane contact sites between the ER and endosomes have been shown to increase as endosomes mature[58], to define endosome fission[11], and to control association of endosomes with the cytoskeleton[15], all of which are essential for

proper endolysosomal function. The TMEM16K proteomics dataset contained multiple proteins known to function at ER-endosomal contact sites including VAPA and VAPB[15,59,60], SNX1 and SNX2[15], Rab7A[58,60,61], and PTP1B[16] (Fig. 2b; Supplementary Data 1), suggesting that TMEM16K acts at or in the proximity of these membrane contact sites.

Hence, we applied in cell culture the same proximity biotinylation approach labeling within a 10-nm range used for proteomics, in order to visualize TMEM16K-dependent labeling of Rab7-positive endosomes (Fig. 5a). We found that TMEM16K-proximity dependent labeling overlapped with endogenous Rab7. For a dynamic view, we performed live imaging of fluorescently

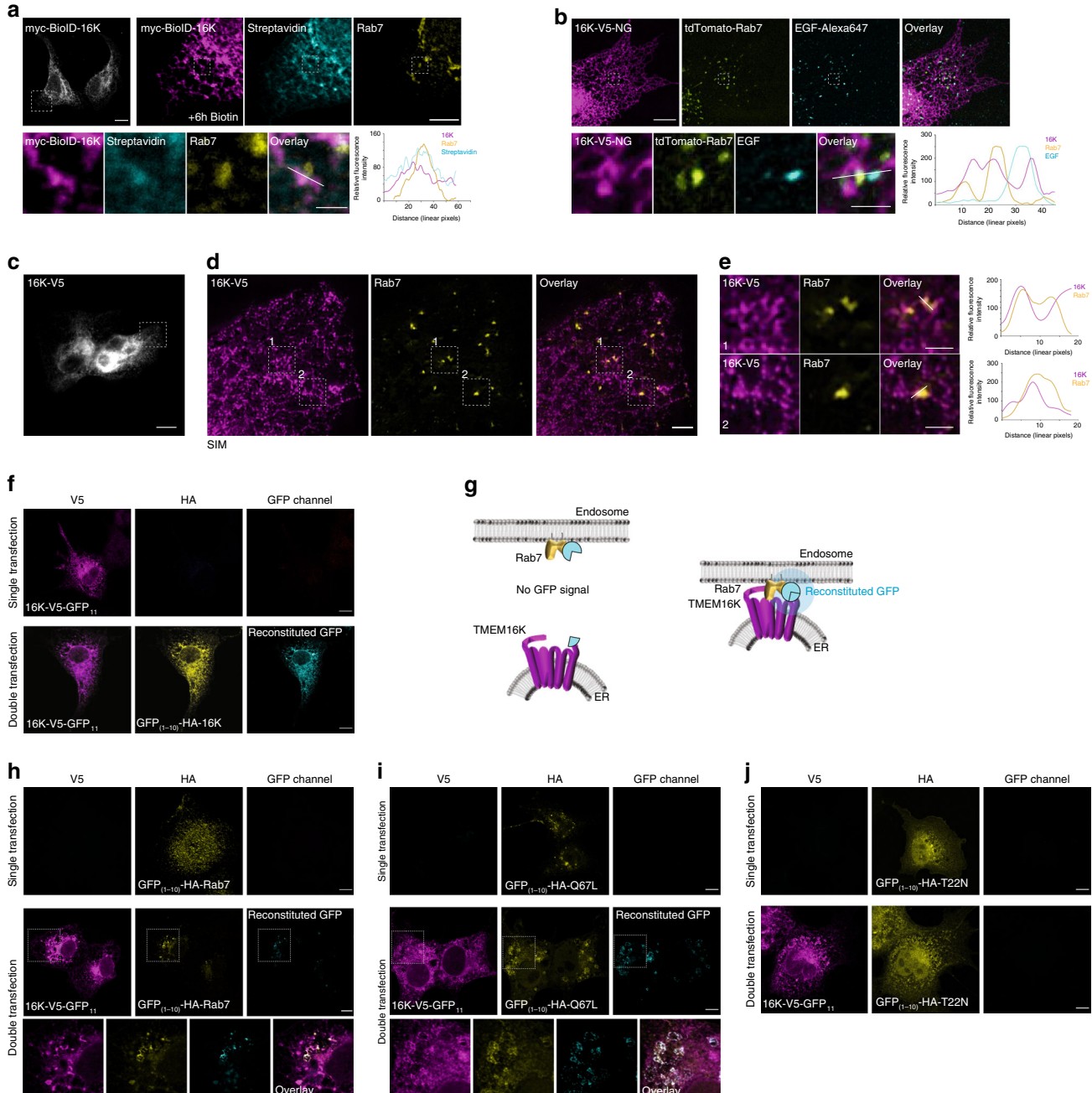

**Fig. 5 TMEM16K at ER-endosome membrane contact sites. a** Immunocytochemistry to visualize TMEM16K proximity labeling of endosomes. COS-7 cells were transfected with TMEM16K tagged with proximity biotinylation enzyme, incubated with biotin for 6 h, and immunostained with fluorescently conjugated Streptavidin and antibody against endogenous Rab7. **a**. Row 1, Left: View of the entire cell expressing myc-BioID-TMEM16K. Scale bar 10 μm. **a**. Row 1, Right: Magnified region of the cell showing myc-BioID-TMEM16K, its pattern of proximity biotinylation and endogenous Rab7. Scale bar 5 μm. **a**. Row 2: High magnification insets with line scan quantification of the three channels marked in overlay. Scale bar 1 μm. **b** Live imaging of the U-2OS cells transfected with TMEM16K-V5-mNeonGreen (TMEM16K-V5-mNG), tdTomato-Rab7, and EGF-Alexa647, pulse chased for 45 min, imaged with spinning disk confocal microscope. See Supplementary Movie 3. **b** Row 1: Snapshots of the live imaging showing TMEM16K, Rab7, EGF and their overlay. Scale bar 10 μm. **b**. Row 2, High magnification insets with line scan quantification of the three channels marked in overlay. Scale bar 0.5 μm **c** Widefield image to view entire cell expressing TMEM16K-V5. Scale bar 10 μm. Inset marks cell region imaged with structured illumination microscopy (SIM). **d**. Single plane structured illumination microscopy of U-2OS cells transfected with TMEM16K-V5 and immunolabelled for endogenous Rab7. Scale bar 2 μm. **e** High magnification insets 1 and 2 from SIM images with corresponding line scan quantification of the two channels marked in overlay. Scale bar 1 μm. **f** Split-GFP assay positive control with cells transfected with TMEM16K-V5-GFP and GFP(1–10)-HA-TMEM16K, as TMEM16K acts as dimer. Scale bar 10 μm. **g** Scheme of the split-GFP experiment where molecule of the GFP can be reconstituted only if the two proteins contact. **h** Split GFP reconstitution assay between TMEM16K-V5-GFP and GFP(1–10)-HA-Rab7. Scale bar 10 μm, inset 2 μm. **i** Split-GFP reconstitution assay betweenTMEM16K-V5-GFP11 and constitutively active mutant of Rab7 Q67L tagged with GFP(1–10) Scale bar 10 μm, inset 2 μm. **j** Split-GFP reconstitution assay between TMEM16K-V5-GFP11 and inactive mutant of Rab7 T22N tagged with GFP(1–10) Scale bar 10 μm.

labeled TMEM16K in the ER, along with fluorescently labeled Rab7 and fluorescently conjugated EGF in endosomes (Supplementary Movie 3, Fig. 5b). These experiments revealed highly dynamic movements of both compartments, as well as their contacts. Using structured illumination microscopy (SIM), we imaged TMEM16K and Rab7-positive endosomes, and visualized dually labeled ER-endosome contact sites (Fig. 5c–e). Next, we used the split-GFP system to specifically evaluate TMEM16K interorganelle contact sites. Split-GFP reconstitution has been extensively used to detect interorganelle contact sites[37–39]. TMEM16K was tagged with a GFP[11] fragment, while several ER (VAPA, OSBPL8) and endosomal (Rab7, OSBPL9, OSBPL11, VPS26, VPS35, SNX1, SNX2) proteins were tagged with the GFP[1-10] fragment. We selected proteins that were identified in the TMEM16K proteomics and implicated in ER-endosomal MCS (VAPA, Rab7, OSBPL11, VPS26, SNX1, SNX2), as well as proteins that are not TMEM16K interaction candidates based on proteomics but are known to participate in similar processes/compartments (OSBPL8, OSBPL9, VPS35), as negative controls. Since TMEM16K forms a dimer[33,56], we validated the split-GFP approach by expressing TMEM16K-GFP[1-10] and GFP[11]-TMEM16K to reconstitute the split-GFP (Fig. 5f). Reconstitution of split-GFP between TMEM16K and any of the tested candidates suggests that they directly interact, bringing the two GFP fragments to such close proximity that they can reconstitute the fluorescent GFP. Inability to reconstitute split-GFP suggests that TMEM16K and the tested candidate are not in close proximity, though we cannot exclude the possibility that steric hindrance may prevent the split-GFP reconstitution (Fig. 5g). Out of all the combinations tested, only Rab7 reconstituted split-GFP with TMEM16K, demonstrating that ER-localized TMEM16K forms contacts with Rab7 endosomes (Fig. 5h; Supplementary Fig. 4). Rab7 is a GTPase that cycles between inactive GDP bound states and active GTP bound states. To evaluate further the specificity of TMEM16K interaction with Rab7, we generated Rab7 mutants: constitutively active Rab7 Q67L mutant that mimics permanently GTP-bound Rab7 and inactive Rab7 T22N mutant that mimics permanently GDP-bound Rab7[62]. We found that TMEM16K was able to reconstitute split-GFP only with the constitutively active Rab7 Q67L mutant (Fig. 5i), but not with the inactive Rab7 T22N mutant (Fig. 5j), further validating the specificity of the observed contact between TMEM16K and Rab7 endosomes.

**TMEM16K N-terminal domain binds endolysosomal phosphatidylinositols.** The yeast TMEM16 protein Ist2p mediates ER-plasma membrane contact sites by directly binding plasma membrane specific phosphatidylinositol-(4, 5)-bisphosphate (PtdIns(4,5)$P_2$) via its C-terminus[63]. The presence of a series of positively-charged residues in the TMEM16K N-terminal cytosolic domain prompted us to hypothesize that, in addition to its interaction with endosomal proteins like Rab7, TMEM16K may directly bind phosphatidylinositols. To test this hypothesis, we purified the N-terminal domains of TMEM16K and two other mammalian family members as control, TMEM16F and TMEM16A, to evaluate their lipid binding via a protein lipid overlay assay (Fig. 6a–c). We found that the N-terminal domain of TMEM16F binds plasphosphatidylinositol-(3,4,5)-phosphate (PtdIns(3,4,5)$P_3$), as recently reported[64]. Unlike TMEM16F and TMEM16A, TMEM16K specifically bound phosphatidylinositols present in endolysosomal compartments, including phosphatidylinositol-3-phosphate (PtdIns3P) (Fig. 6c), the major phosphatidylinositol in endosomes. To evaluate the functional requirement of the N-terminal domain, we generated N-terminal deletion mutant of TMEM16K (Δ1-169 amino acids) (Fig. 6d). Whereas we used the N-terminal 255 amino acids in the protein overlay assay, mutation of the 171th amino acid is causative for

human pathology, so we tested for a truncation mutant that retains this residue. This TMEM16K truncation mutant with N-terminal deletion properly localized to endoplasmic reticulum and could still reconstitute split-GFP with Rab7, demonstrating that the N-terminal domain is dispensable for TMEM16K contacts with endosomes (Fig. 6e). However, with N-terminal deletion the TMEM16K truncation mutant was not able to rescue the endosomal retrograde transport defect of cells from TMEM16K knockout mice, showing that the N-terminal domain is required for TMEM16K function (Fig. 6f, g). This functional requirement is reminiscent of the functional requirement of the binding of the TMEM16F N-terminal domain to plasma membrane phosphatidylinsoitols for the regulation of TMEM16F gating[64]. Our findings suggest that the ER-localized TMEM16K forms contact sites with endosomes, where it binds active GTP-bound Rab7 and endolysosomal phosphatidylinositols like PtdIns3P.

**TMEM16K at ER-endosome MCS.** Next, we sought to address how TMEM16K regulates endosomal function. Like TMEM16K, the VAPA and VAPB proteins, which were detected in the TMEM16K proteomics, act at ER-endosome membrane contact sites and can affect endosomal retrograde trafficking; VAPA and VAPB regulate PtdInsI4P levels on endosomes and subsequent WASH-dependent actin nucleation[15]. If TMEM16K indirectly affects endosomal sorting through this VAPA/B pathway, we would expect it to be associated with similar cellular defects. We first evaluated whether ER-endosome membrane contact sites are globally perturbed in the absence of the TMEM16K. Using electron microscopy[65], we found no difference in the percentage of endosomes in close proximity (~30 nm) to ER between wild type and TMEM16K knockout cells (Supplementary Fig. 5a, b), consistent with the presence of multiple proteins maintaining these contacts[4,66]. We have further strengthened these observations with proximity ligation assay (PLA) in situ, a powerful approach to study contact sites alterations in a quantitative manner. Using VAPB and Rab7 as markers of ER and endosomes, respectively, we found no difference in the extent of ER-endosome MCS as measured by the number of PLA puncta between the wild type and TMEM16K knockout cells (Supplementary Fig. 5c–e), corroborating that the absence of TMEM16K does not globally perturb ER-endosome MCS. As cells lacking VAPA/B accumulate PtdIns4P on endosomes[15] we next assessed PtdIns4P distribution by using the reporter GFP- P4M SidM[67], and found no perturbation in the absence of TMEM16K (Fig. 7a). Likewise, there was no alteration of plasma membrane PtdIns(4,5)$P_2$ visualized with the 2PH-PLCΔ-GFP biosensor or anti-PIP2 antibody (Supplementary Fig. 3a). Furthermore, unlike VAPA/B DKO cells[15], we found no perturbation of the actin cytoskeleton in cells lacking TMEM16K (Fig. 7a). However, by utilizing P40PX-EGFP to detect the distribution of PtdIns3P[68], we found enlarged PtdIns3P vesicles in the absence of TMEM16K (Fig. 7b, d). Reintroducing TMEM16K in KO cells fully rescued the enlarged PtdIns3P vesicles phenotype (Fig. 7b, d). PtdIns3P is a precursor for the generation of PtdIns(3,5)$P_2$ by the only mammalian PtdIns 5-kinase, PIKfyve[69], which when perturbed also leads to enlarged endosomes. Thus, we wondered if this conversion is perturbed in the absence of TMEM16K. As we were not able to reliably visualize PtdIns(3,5)$P_2$ with fluorescent reporter in primary cells from wild type and TMEM16K knockout mice, we used the pharmacological inhibitor of PIKfyve[70], YM201636, and visualized its effects on PtdIns3P (Fig. 7c). Inhibiting PIKfyve in TMEM16K wild type cells recapitulated the TMEM16K knockout phenotype. However, in the TMEM16K knockout cells we did not observe additional cumulative effect, suggesting that conversion of PtdIns3P to PtdIns(3,5)$P_2$[71] is impaired in the absence of

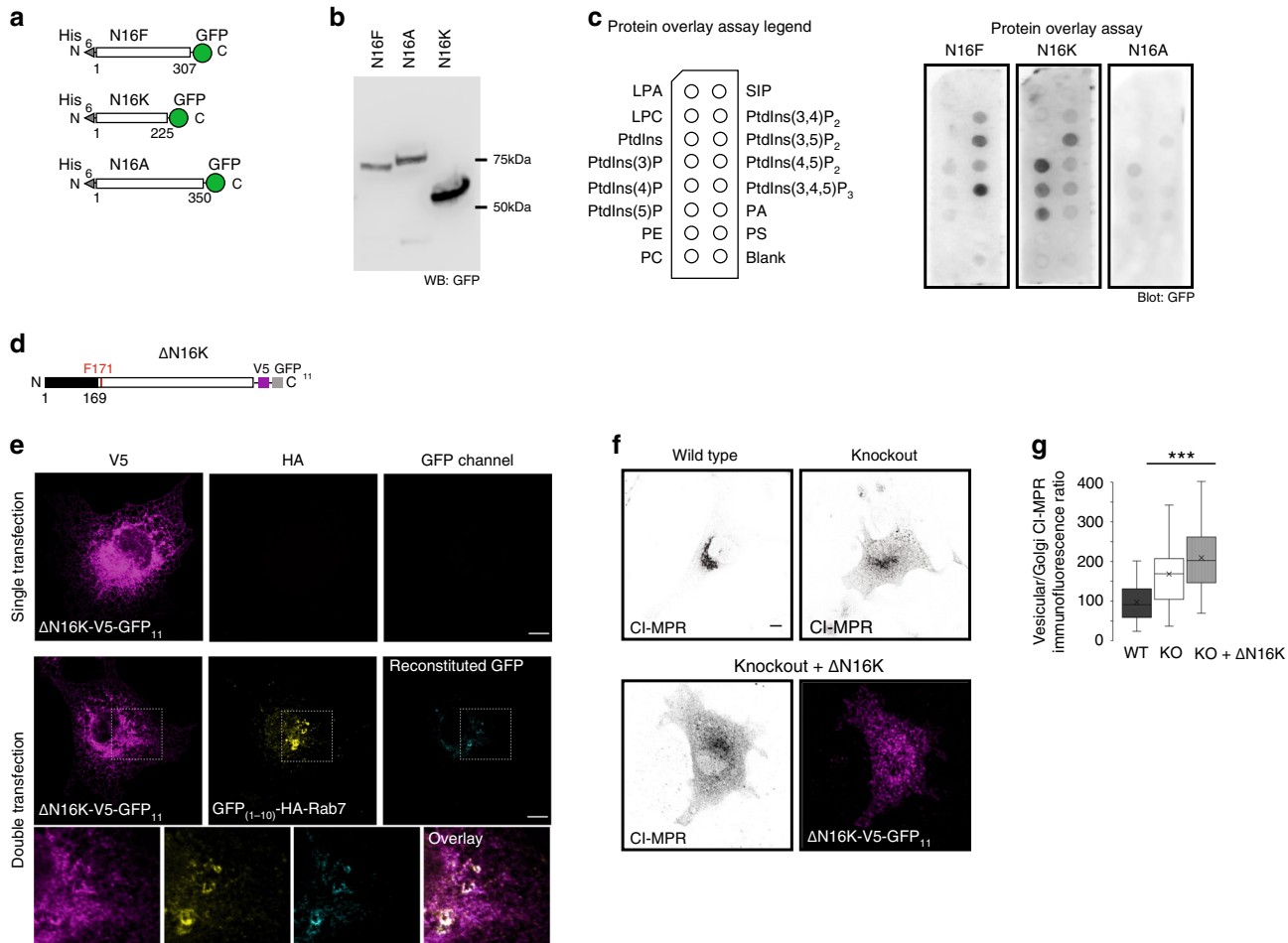

**Fig. 6 TMEM16K N-terminal domain. a** Scheme of purification constructs of N-terminal cytosolic domains of TMEM16F (N16F, predicted MW 63.6 kDa), TMEM16K (N16K, predicted MW 53.4 kDa) and TMEM16A (N16A, predicted MW 67.5 kDa) **b** Western blot of purified N-terminal cytosolic domains revealed with anti-GFP antibody and horseradish peroxidase. **c** Lipid binding of predicted phosphatidylinositol binding domains was evaluated by loading 10 μg of purified proteins to the PIP-strips and visualized with blotting with anti-GFP antibody and horseradish peroxidase. Representative images of three independent blots. **d** Scheme of N-terminal truncation mutant of TMEM16K where 1–169 amino acids were deleted. **e** Split-GFP reconstitution assay to evaluate N-terminal truncation mutant for its ability to reconstitute GFP with Rab7. Cell were single or double transfected cells with ΔN –terminal TMEM16K-V5-GFP$_{11}$ and GFP$_{(1-10)}$ tagged Rab7, Scale bar 10 μm for rows 1, 2, and 5 μm for inset in the row 3. **f** Representative images of the ability of ΔN16K-V5-GFP$_{11}$ to rescue endosomal retrograde trafficking defect when introduced in TMEM16K KO cells as measured by CI-MPR assay described in Fig. 3a (WT and KO from Fig. 3a repeated for clarity of comparison) Scale bar 10 μm. **g** Quantification (same data from Fig. 3a added to this graph for clarity of comparison). Single factor ANOVA, $p$ value = 2.03E−49 with post-test Bonferroni-corrected two sided $t$-test with pairwise comparison with WT (three biological replicates, $n$ = 168 WT, 181 TMEM16K KO, $p$ value = 1.37E−26*** and 161 TMEM16K KO + ΔN16K-V5-GFP$_{11}$ cells, $p$ value = 6.06E−47***). In the box and whiskers plot, the box includes the first quartile and the third quartile, with the central line representing the median. Whiskers represent the minimum and maximum values of data. X inside the box represents the mean of data. Source data are provided as a Source Data file.

TMEM16K (Fig. 7c, d). Altogether, our results strongly indicate that, while there could be coordination of TMEM16K and VAPA/B MCS functions in mediating endosomal retrograde trafficking, TMEM16K affects endosomal sorting in a manner independent of the VAPA/B pathway.

**Lipid scrambling activity of TMEM16K.** Given that N-terminal TMEM16K truncation mutant can reconstitute split-GFP with Rab7, but is unable to rescue the CI-MPR retrograde trafficking defect in TMEM16K knockout cells (Fig. 6d–g), indicates that proximity to endosomes is required, but not sufficient for TMEM16K cellular function. TMEM16K was recently demonstrated to possess calcium regulated phospholipid scramblase activity[33,72,73], translocating phospholipids bidirectionally down their concentration gradients. Therefore, we wondered whether TMEM16K-mediated lipid scrambling function is required for

endosomal sorting. Grafting the 35 amino acids constituting the minimal scrambling domain (SCRD)[74] of the TMEM16F scramblase (Fig. 8a, b) onto the TMEM16A calcium-activated chloride channel conveyed scrambling activity to the chimera. Similarly, grafting SCRD homology region of TMEM16K onto TMEM16A conveyed scrambling activity to the chimera, while grafting SCRD homology region of non-scramblase family members did not convert TMEM16A into a scramblase[72]. To evaluate whether TMEM16K scrambling function is required for endosomal retrograde trafficking, we used the same established approach and generated chimeras substituting the minimal SCRD of TMEM16K with that of non-scramblase TMEM16A or scramblase TMEM16F (Fig. 8a, b). Both TMEM16K-SCRD16A and TMEM16K-SCRD16F chimeras can be efficiently expressed and correctly localized to endoplasmic reticulum (Supplementary Fig. S6). When reintroduced into TMEM16K KO cells, the

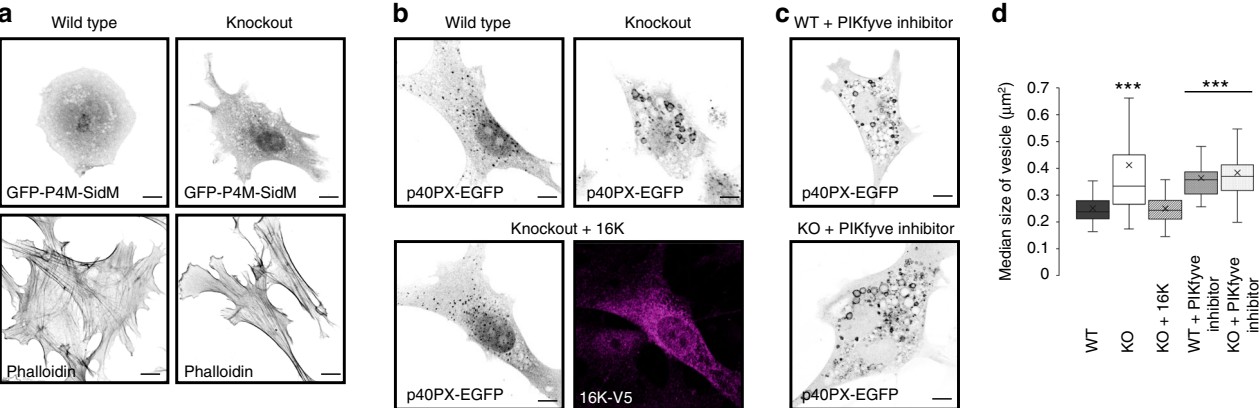

**Fig. 7 Endolysosomal phosphatidylinositols. a** Confocal images of WT and TMEM16K KO cells transfected with GFP-P4M-SidM, biosensor for PtdIns4P or labeled with fluorescence conjugated phalloidin which specifically labels actin network. Representative images from two independent experiments. Scale bar 10 μm. **b**. Confocal images of WT, TMEM16K KO cells and KO cells with reintroduced TMEM16K cells transfected with PtdIns3P biosensor P40PX-EGFP. Scale bar 10 μm. **c**. Confocal images of WT and TMEM16K KO cells transfected with PtdIns3P biosensor P40PX-EGFP and treated with PIKfyve kinase inhibitor YM201636. Scale bar 10 μm. **d**. Quantification of the median size of the PtdIns3P positive vesicles per cell visualized with P40PX-EGFP from 3 biological replicates in the WT, TMEM16K KO and TMEM16K KO cells transfected with wild type TMEM16K cDNA, as well as WT and TMEM16K KO cell treated with YM201636. Single factor ANOVA, $p$ value $= 6.82E-63$ with post-test Bonferroni-corrected two sided $t$-test with pairwise comparison with WT. ($n = 131$ WT, $n = 131$ KO, $p$ value $= 1.50E-18^{***}$, $n = 153$ KO $+ 16$K, $n = 150$ WT $+$ PIKfyve inhibitor, $p$ value $= 5.86E-42^{***}$, $n = 156$ KO $+$ PIKfyve inhibitor, $p$ value $= 1.59E-41^{***}$). In the box and whiskers plot, the box includes the first quartile and the third quartile, with the central line representing the median. Whiskers represent the minimum and maximum values of data. X inside the box represents the mean of data. Source data are provided as a Source Data file.

putative non-scramblase chimera TMEM16K-SCRD16A failed to reconstitute split-GFP with Rab7 (Fig. 8c) and it could not rescue the retrograde trafficking defect as revealed by CI-MPR internalized antibody distribution (Fig. 8e, f). In contrast, the putative scramblase chimera TMEM16K-SCRD16F was able to reconstitute split-GFP with Rab7 (Fig. 8c) and rescue the endosomal retrograde trafficking defect (Fig. 8e, f), suggesting that the lipid scrambling activity of TMEM16K could be required for endosomal sorting.

**Human disease variants**. To better define human mutations linked to spinocerebellar ataxia (SCAR10), we tested three known single amino acid missense mutations (Phe171Ser, Phe337Val and Asp615Asn; Fig. 8a, b)[33,36,75,76]. Based on the crystal structure Phe171Ser and Phe337Val mutations are predicted to interfere with conformation changes of the protein during scrambling, while Asp615Asn lies in the Ca2+-binding site at the dimer interfaces and does not have perturbed lipid scrambling activity in vitro[33]. These human disease variants expressed normally and localized to the endoplasmic reticulum in heterologous cells (Supplementary Fig. 6). Out of these three, Phe171Ser was able to reconstitute split-GFP with Rab7 (Fig. 8d). Remarkably, unlike wild type TMEM16K that was able to completely rescue the CI-MPR retrograde trafficking defect in TMEM16K knockout cells, all these three human disease variants were unable to rescue (Fig. 8e, f). Thus, our results suggest that impairment of the endosomal pathway could be a contributing factor in the development of the SCAR10 pathology.

**Discussion**
The endolysosomal pathway is a series of organelles with a challenging task of internalizing, and properly sorting for recycling, reuse, or degradation of various cargo molecules required for normal cellular function. Our study finds that endosomal sorting is regulated by the lipid scramblase TMEM16K at ER-endosome contact sites. Endoplasmic reticulum-localized TMEM16K forms contacts with endosomes by binding endosomal GTPase Rab7 and

endolysosomal phosphatidylinositols. Loss of TMEM16K leads to dysfunction of endosomal sorting, which could be rescued by wild type TMEM16K but not mutant TMEM16K bearing the human disease point mutations causing spinocerebellar ataxia. The defect in the later stages of the endolysosomal pathway could be caused by mistargeting of proteins needed for the late endosome/lysosome function due to defects in endosome sorting in the absence of TMEM16K[71]. In addition to the observed cellular defects, we found progressive deterioration of the neuromuscular function in the TMEM16K knockout mice. Dysfunctions in endosomal sorting are known to accumulate over the lifetime and represent a converging mechanism shared by a broad range of neurodegenerative diseases[47,77], and our findings opens new avenues how cells manage endosomal sorting.

Our results are consistent with a model in which the TMEM16K phospholipid scrambling function could, upon TMEM16K activation by binding of the specific phosphatidylinositols[64,78] and calcium[33,56], selectively modulate the local lipid environment in the endoplasmic reticulum at these sites of contact. Distribution of phospholipids across the ER membrane is thought to be largely symmetrical, with the exception of phosphatidylserine[73,79], and TMEM16K was shown to be required for its calcium-induced leaflet redistribution[73]. A number of recent studies have highlighted the importance of lipid microdomains in protein sorting[80]. For example, translocation of phosphatidylserine across leaflets is required for sorting at the trans-Golgi complex in yeast[81], and loss of phosphatidylserine asymmetry impairs sorting of early endosomes in C. elegans[82]. Just how modulating leaflet composition in the ER would affect endosomal sorting is an open question. TMEM16K-modulated lipid availability in the ER at the sites of contact with endosomes could locally recruit proteins[83], modulate local protein activity, or promote direct transfer of lipids between ER and endosomes[25], providing necessary cues for endosomal sorting. Interestingly, the TMEM16K yeast homolog Ist2p was recently shown to support transport of phosphatidylserine from ER to the PM[32] through interaction with the lipid transfer protein Osh6. TMEM16K proteomics includes lipid transfer protein OSBP-related protein 11 (OSBPL11)[84] from the same family.

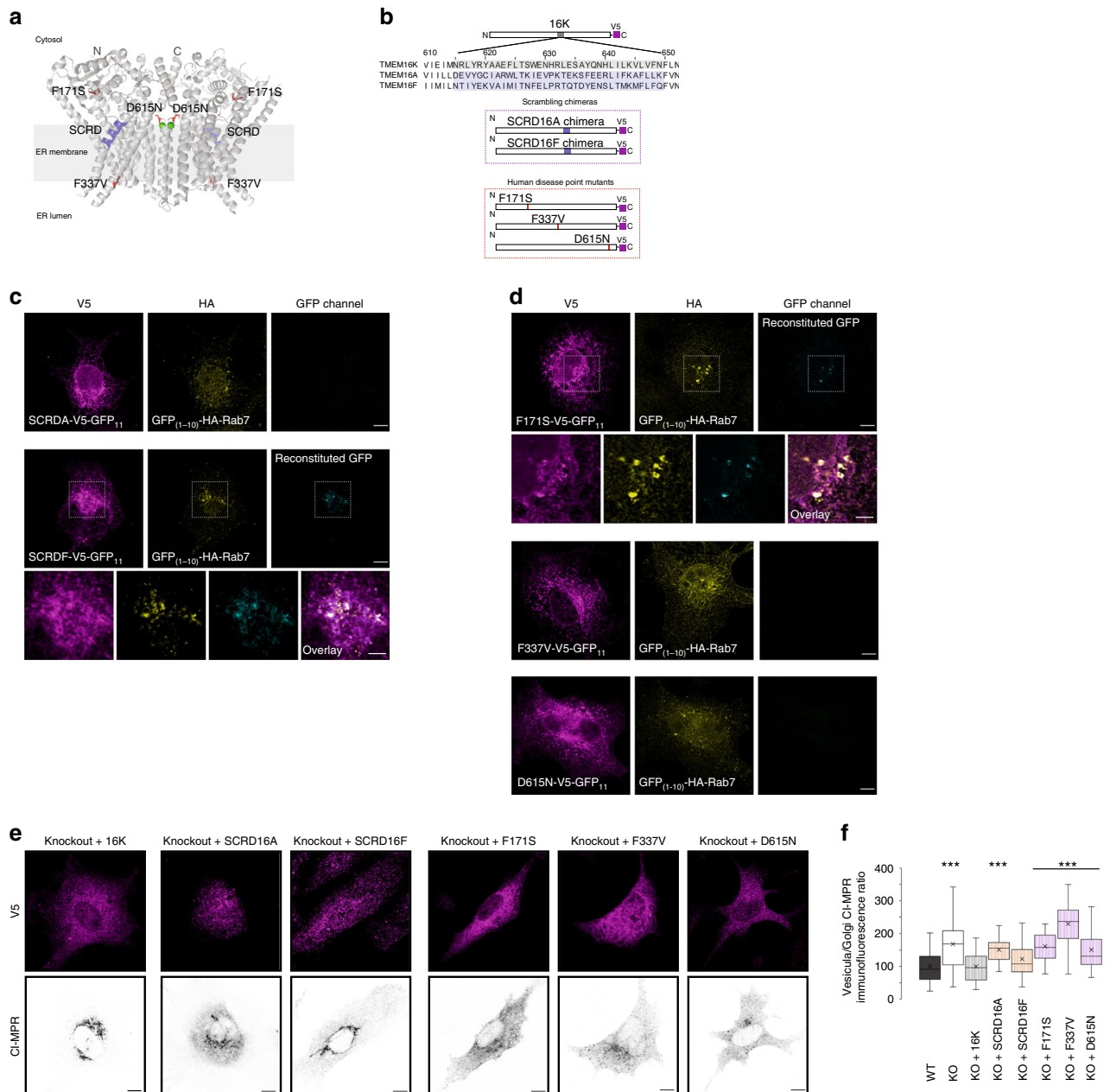

**Fig. 8 TMEM16K scrambling domain chimeras and human disease variants. a** Topology of TMEM16K dimer in the endoplasmic reticulum membrane with labeled human disease point mutations and SCRD (minimal lipid scrambling domain) based on published crystal structure[33] and represented in Pymol. **b** Scheme of the SCRD chimeras and human disease point mutation constructs **c** Split-GFP reconstitution assay of scrambling domain chimeras (SCRDA, SCRDF) tagged with GFP$_{11}$ and GFP$_{(1-10)}$ tagged Rab7. Scale bar 10 μm, inset 5 μm. **d** Split-GFP reconstitution assay of human point mutants (F171S, F337V, D615N) tagged with GFP$_{11}$ with GFP$_{(1-10)}$ tagged Rab7. Scale bar 10 μm, inset 5 μm. **e** Representative images of the ability of SCRD chimeras and human disease point mutants to rescue endosomal retrograde trafficking defect when introduced in TMEM16K KO (KO + 16K cDNA from Fig. 3a repeated for clarity of comparison) Scale bar 10 μm. **f** Quantification done as in Fig. 3a (same data added to this graph for clarity of comparison). Single factor ANOVA, p value = 2.1E−117 with post-test Bonferroni-corrected two sided t-test with pairwise comparison with WT, (three biological replicates, n = 168 WT, 181 TMEM16K KO, p value = 1.37E−26*** and 134 KO + 16K cells, 130 KO + 16KSCRD16A, p value = 5.05E−29***, 145 KO + 16KSCRD16F, n = 145 KO + F171S, p value = 1.98E−36***, 138 KO + F337V, p value = 1.37E−17***, 155 KO + D615N cells, p value=1.37E-29***). In the box and whiskers plot, the box includes the first quartile and the third quartile, with the central line representing the median. Whiskers represent the minimum and maximum values of data. X inside the box represents the mean of data. Source data are provided as a Source Data file.

Altogether, our results open the possibility that endosomal sorting could be enabled by modulating the lipid environment in trans at sites of contact between organelles, an intriguing hypothesis to be evaluated in future studies.

Our study further raises the possibility that mammalian TMEM16 family members could have functional roles as interorganelle regulators. Indeed, mammalian TMEM16H was recently shown to regulate Ca$^{2+}$ signaling at the ER and plasma membrane MCS[85]. Considering that mammalian TMEM16K and other family members are important modulators of cellular physiology and human pathology, determining whether and how they are involved in interorganelle communication would

significantly improve our understanding of this protein super-family as well as cellular communicome, the cellular communication network.

## Methods

**Antibodies and chemicals**. We used following primary antibodies: rabbit anti-V5 (Cell Signaling tech, #13202, 1/1000), mouse anti-V5 mouse (Invitrogen, # R960-25, 1/1000), mouse anti-FLAG (Sigma, # F1804, 1/1000), rabbit anti-GM130 (Abcam, # ab52649, 1/500), rabbit anti-Rab7 (Cell Signaling tech # 9367, 1/500), mouse anti-myc (kind gift from J. Michael Bishop, clone 9E10, 1/1000), rat anti-HA (Roche, # 11867431001, 1/1000), mouse anti-GFP (Roche, # 11814460001, 1/1000), rabbit anti-calreticulin (Abcam, # ab2907, 1/1000), mouse anti-PDI (Abcam, # ab2792, 1/500), rabbit anti-giantin (kind gift from Marc Von Zastrow, 1/1500), mouse anti-Golgin97 (Molecular probes, # A21270, 1/200), rabbit anti-mCherry (Abcam, # ab167453, 1/1000), chicken anti-mCherry (Novus Biologicals, #NBP2-25158, 1/200), mouse anti-EEA1 (BD Biosciences, #610456, 1/200), mouse anti-PIP2 (Abcam, # ab11039, 1/250), rabbit anti-TGN38 (Novus Biologicals, #NBP1-03495SS, 1/100), and mouse anti-VAPB (R&D systems, #MAB7329-SP, 1/100 for IF and PLA in situ). Secondary antibodies (used at 1/400 for IF and 1/2000 for HRP revelation of Western blots and protein overlay assay) and Strep-tavidin conjugated with Alexa Fluor 647 (# 016-600-084, 1/2000) were purchased from Jackson laboratories or Invitrogen. Phalloidin conjugated to Alexa Fluor 633 was obtained from Molecular probes (# A-22284, 1/400). Cholera Toxin Subunit B, AlexaFluor 555 conjugate was purchased form ThermoFisher (# C22843). α-Bungarotoxin, Alexa Fluor 488 conjugate (# B13422), Lysosensor Green DNP-189 (#L-7535), Alexa Fluor 647 conjugated EGF (#E-35351) and Alexa Fluor 647 conjugated transferrin (#T-23366) were purchased from Life Technologies. FCCP (#C2920-10MG) was purchased from Sigma.

**Plasmids**. We obtained mouse TMEM16K cDNA from Genscript (Clone ID: OMu10422D, pCDNA3.1-TMEM16K-FLAG). Myc-BioID and BioID-HA were a gift from Kyle Roux (Addgene # 35700 and # 36047, respectively). Flag-CIMPR was a gift from Marc Von Zastrow, UCSF. CIBN-CAAX was a gift from Pietro de Camilli, Yale. GFP-P4M-SidM was a gift from Tamas Balla (Addgene, # 51469). p40PX-EGFP was a gift from Michael Yaffe (Addgene, #19010). pKanCMV-mClover3-mRuby3 was a gift from Michael Lin (Addgene # 74252). pSBtet-RB was a gift from Eric Kowarz (Addgene # 60506). pcDNA3.1-GFP(1-10) was a gift from Bo Huang (Addgene plasmid # 70219). 2PH-PLCdelta-GFP was a gift from Sergio Grinstein (Addgene, # 35142). pEGFP-N1-VAPA was a gift from Axel Brunger (Addgene # 18874). tdTomato-Rab7, ER-tdTomato and mNeonGreen-mRuby2-FRET-10 were obtained from UCSF Nikon Imaging Center Library. We have obtained cDNA for VPS26A (Clone ID: BC022505) OSBPL9 (Clone ID: BC025978) and OSBPL11 (Clone ID: pCS6(BC065213)) from Transomics. We have received SNX1 cDNA as a gift from Ewan Reid, University of Cambridge. SNX2 and VPS35 cDNA were gift from Marcel Verges, Universitat de Girona. ORP8 was gift from Francesca Giordano, Institut de Biologie Integrative, Gif-sur-Yvette. mCherry-tagged GPI RUSH construct was gift from Franck Perez, Institut Curie. mCherry-tagged Transferrin Receptor (TfR) RUSH construct and GFP-tagged CD-MPR RUSH construct were gift from Juan Bonifacino, NIH.

Detailed list of all the primers as well as description of all the constructs used and designed for this study is given in Supplementary Table 1. Site-directed mutagenesis was performed using PfuTurbo polymerase (Agilent Technologies) followed by DpnI digestion protocol. All other plasmids were constructed using classical subcloning or Gibson assembly using Hot Start High Fidelity Q5 polymerase from NEB. All constructs generated and used in this study were verified with sequencing of the coding region of the plasmids.

**TMEM16K mice models**. We obtained TMEM16K conditional knockout mice (Ano10tm1a(EUCOMM)Wtsi)) generated by the International Mouse Knockout Consortium and ordered from EMMA (EMMA ID: 08927). Ubiquitous TMEM16K knockout mice were generated by crossing with the actin-driven Cre line, while crossing with nestin-Cre line generated neuron specific TMEM16K knockouts. All procedures performed have been approved by the UCSF Institutional Animal Care and Use Committee (IACUC). Wild type allele is identified with Ano10_257998_F (CACTCCCTCATCCCATTCTTG), and Ano10_257998_R (AGACGGCCACCT TACCACAG) primers (band size 433 bp). Mutant allele is detected with PCR with Ano10_257998_F and CAS_R1_Term (TCGTGGTATCGTTATGCGCC) primers (band size 156 bp). Animal cohorts were generated from heterozygote breedings, where WT and TMEM16K knockout littermates were kept gender segregated but genetically mixed with up to five animals per cage. Mice were housed for up to 24 months in the Laboratory Animal Resource Center in Rock Hall, Mission Bay Campus. Mice had access to food and water ad libidum, with provided additional enrichment in the cages and weekly changes of the bedding. Animal health was monitored by the UCSF IACUC staff. In addition, rodent health in containment barriers was monitored through an ongoing Sentinel Program. UCSF has an approved Assurance of Compliance with the U.S. Public Health Service Policy on Humane Care and Use of Laboratory Animals by Awardee Institutions (#3400-01) on file with the Office of Protection from Research Risks, NIH.

**Neuromuscular junction staining**. Animals were anesthetizes with isofluorane, and perfused first with PBS and then with 4% PFA. In all the animals, we have dissected, analyzed and compared the same muscle localized in the hindlimbs. After isolation, muscles were immersed in blocking buffer containing 0.5% Triton-X 100 and 10% donkey serum in PBS at 4 °C overnight. The muscles were then washed three times with 1× PBS at room temperature for 15 min each time followed by incubation with fluorescently labeled α-Bungarotoxin overnight at 4 °C on a rotating platform. The muscles are again washed three times with 1x PBS, and mounted Fluoromount-G medium (SouthernBiotech) with weight applied on the cover glass to obtain flat preparation suitable for imaging. NMJ were imaged on confocal microscope as z-stacks and represented as maximum intensity projections to ensure entire structure is captured, and then analyzed in Fiji Software (ImageJ, NIH).

**Hindlimb clasping and ledge assays**. Behavior of WT and TMEM16K KO littermates was analyzed using hindlimb clasping and ledge assays. Hindlimb clasping is a marker of disease progression in a number of mouse models of neurodegeneration[86], while ledge assay is a direct measure of coordination, which is impaired in cerebellar ataxias and many other neurodegenerative disorders. The evaluation of mice was done in a following manner[38]: Each measure is recorded on a scale of 0–3 depending on the severity of phenotype. For hindlimb clasping, mouse was lifted clear of all surrounding objects by grasping the tail near its base. The hindlimb position was observed for 1 min. If the hindlimbs were consistently splayed outward, away from the abdomen, it was assigned a score of 0. If one hindlimb retracted toward the abdomen for more than 50% of the time suspended, it received a score of 1. If both hindlimbs were partially retracted toward the abdomen for more than 50% of the time suspended, it received a score of 2. If its hindlimbs were entirely retracted and touching the abdomen for more than 50% of the time suspended, it received a score of 3. For the ledge assay, mouse was observed as it walked along the cage ledge and lowered itself into its cage. A wild type mouse walks along the ledge without losing its balance, and lowers itself back into the cage using its paws. This was assigned a score of 0. If the mouse lost its footing while walking along the ledge, it received a score of 1. If it did not effectively use its hind legs, or landed on its head rather than its paws when descending into the cage, it received a score of 2. If it fell off the ledge, while walking or attempting to lower itself, or shaked, it received a score of 3. All behavior analysis was done blinded of the mice genotype, with each mice tracked during the analysis with a random numeric code.

**RNA extraction and RT-PCR**. Animals were euthanized with $CO_2$ and tissue was immediately dissected on ice. Total RNA was extracted with TRIzol (ThermoFisher) and first-strand cDNA was synthesized with SuperScript™ III First-Strand Synthesis System kit (ThermoFisher) or High Capacity cDNA Reverse Transcription Kit (Applied Biosystems). We quantified the cDNA with NanoDrop and performed PCRs with Q5 Hot Start High Fidelity polymerase (NEB) using equal amounts of obtained cDNA from each tissue sample to detect presence of TMEM16K (primer set 1: CATGGCCATCATTGGACTGCCC, GCACAGCCACGCTTCCACAC, size 126 bp; primer set 2: GCCATGCGGGCCTTCACCTA, CAGTCCAATGATGGCC ATGGGG, size 405 bp) and housekeeping genes Gadph (primers: TGGCCCCTCT GGAAAGCTGTG, AGTTGGGATAGGGCCTCTCTTGC, size 501 bp) and β-actin (primers: ATGAGCTGCGTGTGGCCCCTG, GACGCAGGATGGCGTGAGGG, 258 bp). DNA Ladder used was GeneRuler 1 kb Plus DNA Ladder from ThermoScientific.

**Cell culture and transfection**. Primary mouse embryonic fibroblast (MEF) cell cultures were generated from 13.5–14.5 days old embryos obtained from time pregnancies set up from heterozygote breedings. Each embryo was genotyped and individually processed for primary culture. Primary MEF culture was established using standard protocol. Primary MEF were used for maximum of five passages. HEK293, COS-7, U-2OS and primary MEF cells were cultured in Dulbecco's modified Eagle's medium (DMEM) containing 10% fetal bovine serum (FBS) and 1% penicillin/streptomycin at 37 °C and 5% $CO_2$.

We used HEK293 cells for proteomics due to their easy maintenance, suitability for scaling needed for biochemistry, and human origin to simplify peptide detection. COS-7 cells were used for imaging due to their flat morphology, ease of maintenance and transfectability. U-2OS cells were used for colocalization analysis due to their human origin suitable for larger number of antibodies and suitable flat morphology.

Transfection of plasmids into HEK293, U-2OS and COS-7 was carried out with Lipofectamine 3000 (Life Technologies), Fugene6 (Promega) or Jetprime (Polyplus transfection) following manufacturer's instructions. Primary MEF were electroporated with the Amaxa Nucleofector using Mouse/Rat Hepatocyte Nucleofector™ Kit (#VPL-1004) according to manufacturer's instructions.

The amounts of the DNA transfected were per 24-well coverslip: 20 ng for all Rab7 constructs (WT Rab7, T22L, Q67N), and 500 ng for all other constructs. For electroporation we used 2 μg of DNA per construct for $2 \times 10^6$ cells.

**Immunofluorescence**. Cells were fixed for 15 min with 4% PFA; quenched for 30 min in autofluorescence reducing solution (50 mM $NH_4Cl$ in PBS); and blocked

with 1x PBS/5% normal donkey serum/0.3% Triton X-100 for 30 min. Primary antibodies were incubated overnight at 4 °C in 1x PBS/1% BSA/0.3% Triton X-100. After three washes, secondary antibodies were incubated for 1 h at room temperature before mounting in Fluoromount-G medium (SouthernBiotech) for immunochemistry, or Vectashield for 3D-SIM.

**Conventional microscopy**. We performed majority of the fixed and live imaging on Leica SP8-X inverted confocal system equipped with HyD hybrid detectors, adaptive focus control and Okolab environmental incubator cage. Live imaging was performed on Nikon Ti inverted fluorescence microscope with CSU-22 spinning disk confocal, EMCCD camera, and incubator, $CO_2$, and humidity control at the UCSF Nikon Imaging Center. Images are represented using pseudocolors suitable for color-blind palette.

**Evaluation of Golgi complex morphology**. We analyzed Golgi complex morphology by performing immunochemistry on wild type and TMEM16K knockout primary cells with either antibody against TGN38 as trans-Golgi complex marker, or GM130 as cis-Golgi marker. Z-stacks image series were acquired on a Leica SP8-X confocal microscope, with a pinhole of 0.5 AU and voxel depth of 0.19 μm. Imaris software (Oxford Instruments) was used to reconstruct Golgi complex and quantify volume and area for each Golgi complex. Index of fragmentation is defined as ratio of volume and area.

**Proximity ligation assay (PLA) for in situ detection of ER-endosome contacts**. Mouse wild type and TMEM16K knockout fibroblasts were seeded on poly-L-lysine-coated 8-well chamber slide at density of 20,000 cells per chamber, fixed with 4% PFA for 10 min, and subjected to proximity ligation assay according to manufacturer's protocol (SigmaAldrich, Duolink® In Situ Detection Reagents FarRed, #DUO92013). Briefly, after permeabilization with 1x PBS/1% BSA/0.3% Triton X-100 for 30 min, the cells were subjected to blocking, incubation with mouse anti-VAPB and rabbit anti-Rab7 antibodies overnight at 4 °C, hybridization with PLA probes, ligation, amplification, and mounted in Duolink mounting media with DAPI (SigmaAldrich, #DUO82040-5ML). Same procedure was done without inclusion of primary antibodies, as negative control to confirm the specificity of the observed PLA signal. Nonoverlapping images were randomly acquired throughout the slide of each sample on the Leica SP8-X confocal microscope. The Fiji Software (ImageJ, NIH) was used to quantify the number of PLA puncta (https://fiji.sc) indicative of a close apposition between VAPB (ER) and Rab7 (endosome). The number of PLA puncta measured per image was expressed as ratio to number of nuclei in the same image, giving a measurement of the average number of puncta per cell.

**Electron microscopy**. Primary wild type and TMEM16K knockout cells were fixed with 4% paraformaldehyde (PFA) followed by post-fixation in EM fixative (2% PFA and 2.5% glutaraldehyde in 0.1 M phosphate buffer, pH = 7.4) and processed for electron microscopy at the UCSF Veterans Affairs Medical Center Pathology Core. Electron micrographs were analyzed blinded, where endosomes and endoplasmic reticulum were separately identified and labeled using Fiji Software (ImageJ, NIH). The percentage of endosomes with an ER contact site were quantified as described previously[65], where endosome-ER membrane contacts were defined as proximity >30 nm.

**Structured illumination microscopy (SIM)**. SIM super resolution imaging was performed on DeltaVision OMX SR imaging system at the UCSF Nikon Imaging Center. We transfected U-2OS cells with TMEM16K-V5, replated 6 h later on high precision coverslips suitable for high-performance microscopy (Paul Marienfeld GmbH & Co, # 0117520, 1.5H, 12 mm ø), immunostained 24 h later with mouse anti-V5, rabbit anti-Rab7 antibodies, and corresponding secondary antibodies, and mounted in Vectashield. Images are represented using pseudocolors suitable for color-blind palette.

**BioID proximity biotinylation**. To test whether the TMEM16K BioID-fusion proteins were enzymatically active and capable of biotinylating, HEK293 cells were transfected with myc-BioID-TMEM16K or TMEM16K-BioID-HA. Biotin was added (50 μM) for 1 h or overnight, cells were fixed, and biotinylation was visualized with a streptavidin probe conjugated to a fluorescent label.
To purify protein complexes surrounding TMEM16K via in situ BioID-catalyzed biotin labeling, we transfected HEK293 cells with BioID-myc-TMEM16K or TMEM16K-HA-BioID and incubated with 50 μM biotin overnight (12–16 h). Mock transfected cells were processed in parallel to account for endogenously biotinylated proteins. Cells were washed 3× with ice cold PBS and lysed with lysis buffer with protease inhibitors (Roche). Cell lysate was centrifuged for 30 min at 15,000 g and supernatant was applied to streptavidin-conjugated magnetic beads (Dynabeads® MyOne™ Streptavidin C1, Invitrogen, # 65001). Beads were extensively washed 5× with lysis buffer followed by 5× washes with ice-cold PBS, and flash frozen in liquid nitrogen and stored at −80 °C for mass spectrometry. The experiment was performed in a minimum of three biological replicates per conditions.

**Mass spectrometry**. Sample or control-incubated streptavidin magnetic beads were resuspended in 5 mM DTT in 100 mM $NH_4HCO_3$ and incubated for 30 min at room temperature. After this, iodoacetamide was added to a final concentration of 7.5 mM and samples incubated for additional 30 min. 0.5 μg of sequencing grade trypsin (Promega) was added to each sample and incubated at 37 °C overnight. Supernatants of the beads were recovered, and beads digested again using 0.5 μg trypsin in 100 mM $NH_4HCO_3$ for 2 h. Peptides from both consecutive digestions were recovered by solid phase extraction using C18 ZipTips (Millipore), and resuspended in 0.1% formic acid for analysis by LC-MS/MS. Peptides resulting from trypsinization were analyzed on a QExactive Plus (Thermo Scientific), connected to a NanoAcquity™ Ultra Performance UPLC system (Waters). A 15-cm EasySpray C18 column (Thermo Scientific) was used to resolve peptides (90-min 2–30% gradient with 0.1% formic acid in water as mobile phase A and 0.1% formic acid in acetonitrile as mobile phase B. MS was operated in data-dependent mode to automatically switch between MS and MS/MS. The top ten precursor ions with a charge state of 2+ or higher were fragmented by HCD. Peak lists were generated using PAVA in-house software[87]. All generated peak lists were searched against the human and mouse subsets of the SwissProt database (SwissProt.2015.12.1) (plus the corresponding randomized sequences to calculate FRD on the searches, and adding sequences for BioID when necessary), using Protein Prospector[88]. The database search was performed with the following parameters: a mass tolerance of 20 ppm for precursor masses; 30 ppm for MS/MS, cysteine carbamidomethylation as a fixed modification and acetylation of the N terminus of the protein, pyroglutamate formation from N terminal glutamine, and oxidation of methionine as variable modifications. All spectra identified as matches to peptides of a given protein were reported, and the number of spectra (Peptide Spectral Matches, PSMs) used for label free quantitation of protein abundance in the samples. The mass spectrometry proteomics data have been deposited to the ProteomeXchange Consortium[89] via the PRIDE[90] partner repository with the dataset identifier PXD018990.

**Proteomic dataset analysis**. TMEM16K proteome candidates list was generated from a minimum of three independent runs per condition. All proteins that had more than one peptide detected in control conditions were eliminated from further analysis. Next, only those proteins that were at least threefold enriched compared to control condition were considered potential interactors. We have visualized obtained candidate list as protein-protein interaction (PPI) network generated in Cytoscape[91] using String database (Confidence score cutoff = 0.7). We next analyzed network parameters with NetworkAnalyzer to obtain centrality measures and mapped size of the nodes to the degree of the node parameter, where higher degree indicates a hub. Functional enrichment of clusters in the TMEM16K PPI network were further identified and quantified (Enrichment $p$ value cutoff = 0.005) with String Functional Enrichment app in the Cytoscape software, where network was visualized with Edge-weight Spring Embedded Layout. Functional enrichment term corresponding to the major identified clusters were color-coded on the network. Major clusters in the TMEM16K protein interaction network were identified with MCODE cluster app in Cytoscape using default settings and represented as simplified network overlayed with corresponding labels of the previously identified functional enrichments.

**CI-MPR assay**. To perform CI-MPR assay, cells were transfected with Flag tagged CI-MPR. 24 h later cells were starved with serum-free DMEM for 6 h. We then incubated cells in serum free DMEM with 1/1000 mouse anti-Flag antibody (Sigma F1804) for 60 min at 37 °C. They were subsequently washed with PBS, fixed with 4% PFA and then stained for the internalized antibodies by immunofluorescence. The imaging was done on confocal microscope and analyzed in Fiji Software (ImageJ, NIH). The fluorescence intensity within a $10 \times 10$ μm$^2$ region centered on the Golgi complex was then measured. The non-Golgi vesicular fluorescence intensity was obtained by measuring the fluorescence intensity in the $10 \times 10$ μm$^2$ region between Golgi and cell periphery. Data are presented as the non-Golgi vesicular/Golgi CI-MPR fluorescence ratio for each cell.

**Cholera toxin subunit B assay**. Cells were incubated with cholera toxin subunit B (CTxB) conjugated with Alexa 555 (stock 1 mg/ml) at 1/1000 dilution in cell culture medium for 3 min at 37 °C. Coverslips were washed and chased for 1 h, washed with PBS and fixed for 15 min with 4% PFA. Cells were immunostained for Golgi marker GM130. Endosomal retrograde trafficking of CTxB was measured as amount of colocalization with the GM130 using Pearsons colocalization coefficient in Fiji Software (ImageJ, NIH).

**Transferrin internalization assay**. For the pulse-chase experiment examining transferrin internalization, WT and TMEM16K KO cells were washed with ice-cold PBS and incubated at 4 °C for 1 h in DMEM containing 25 μg/ml Alexa Fluor 647-conjugated transferrin and 0.1% BSA. Unbound transferrin was removed with 2× wash with cold medium and cells were allowed to internalize the transferrin at 37 °C for 1 h. The cells were subsequently washed with cold acidic buffer (0.2 M 100% acetic acid, 0.5 M, NaCl, pH 4.2) three times to strip surface-residing transferrin and then washed with phosphate-buffered saline and fixed. Transferrin

internalization was measured as fluorescence intensity of internalized transferrin per cell (AU) in Fiji Software (ImageJ, NIH).

**EGF colocalization assay**. After 16 h of serum starvation, WT and TMEM16K KO cells were stimulated with 100 ng/ml of Alexa 647 conjugated EGF for 3 min at 37 °C, and washed to remove EGF from the medium. The cells were fixed at 10, 15, 40, and 60 min after the initial exposure to Alexa 647-EGF, and immunostained with anti-Rab7 antibody. Images were acquired on the confocal microscope and colocalization between the Alexa647-EGF and Rab7 was analyzed in Fiji Software (ImageJ, NIH) using JaCOP plugin.

**RUSH trafficking assay**. To perform the RUSH secretory assay[54,55] WT and TMEM16K KO primary cells were electroporated with mCherry-tagged GPI RUSH construct, mCherry-tagged Transferrin Receptor (TfR) RUSH construct or GFP-tagged CD-MPR RUSH construct. All used RUSH construct used KDEL as streptavidin hook, blocking RUSH constructs in the ER in the biotin-free medium. Biotin-free media was generated by incubating the DMEM-FBS media with streptavidin coupled to magnetic beads for 60 min. Magnetic nature of the beads allowed easy removal from the media, followed by 0.22 μm filtration to ensure media sterility. Upon addition of the biotin, constructs were released from the endoplasmic reticulum in synchronized manner allowing evaluation of their transport through the biosynthetic pathway. Hence, depending on the construct used, cells were fixed at 0, 15, 60, or 120 min after biotin addition to the medium, and visualized with immunochemistry with anti-mCherry or anti-GFP antibody. GPI is transported with a faster dynamic from the ER through the Golgi to the plasma membrane. TfR is transported with a slower dynamic from the ER through the Golgi to the plasma membrane, from which it gets endocytosed and recycled back to the plasma membrane. CD-MPR is transported from the ER to the trans-Golgi, from where it bypasses plasma membrane and gets directly transported to the endosomes. The RUSH assay for different constructs is quantified on cells fixed 60 or 120 min after biotin addition. As GPI is transported to the plasma membrane, we quantified ratio of the surface to the total GPI detected in the cell, where the amount of surface GPI was evaluated by surface staining with rabbit anti-mCherry antibody and expressed as the ratio to total GPI detected with chicken anti-mCherry antibody. In case of TfR, we evaluated Pearsons colocalization coefficient with the early endosomal marker EEA1 using JaCoP plugin in Fiji Software (ImageJ, NIH). CD-MPR transport was likewise evaluated by analyzing colocalization with the trans-Golgi marker GM130, and in comparison to mCherry-TfR RUSH construct.

**Lysosensor assay**. After 16 h of serum starvation, WT and TMEM16K KO cells were loaded for 30 min at 37 °C with 1 μM Lysosensor Green DNP-189 in culture medium, washed 2× with warm live imaging medium and immediately imaged on confocal microscope for up to 3 min per coverslip to ensure comparable time loaded with the dye between experiments. For experiments where protonophore FCCP was used, time-lapse (30 s interval) was acquired and FCCP was added to final concentration of 2 μM at 120 s. Images were analyzed in Fiji Software (ImageJ, NIH) where fluorescence intensity per cell was measured.

**Protein purification**. Recombinant GFP-fusion proteins were purified from BL21 (DE3) cells (NEB, # C2527H) using an N-terminal H₆-tag[42]. Protein expression was induced at 30 °C by adding 1 mM IPTG. After 2.5 h the cells were harvested, resuspended in bacterial lysis buffer (50 mM Tris-HCl pH 7.5, 250 mM NaCl, 2 mM imidazol, 0.5 mM EDTA, 10 mM β-mercaptoethanol) and lysed with a homogenizer. After ultracentrifugation (Beckman 45 Ti rotor, 32,000 rpm = 80,110 g, 45 min, 4 °C) the supernatant was incubated for 1 h at 4 °C with Ni-NTA-agarose (Qiagen), washed several times with lysis buffer and the protein was eluted by incubation for 3 h with lysis buffer containing 500 mM imidazol. Eluted protein was dialyzed using Slide-A-Lyzer™ Dialysis Cassettes, 3.5K MWCO (Thermo-Fisher) against a buffer with 25 mM HEPES-KOH pH 7.4, 250 mM potassium acetate. Protein concentration was determined with Bradford assay, and proteins stored at −80 °C.

**Protein lipid overlay assay**. Overlay assays was performed following manufacturers instructions. Nitrocellulose-immobilized phospholipids (PIP strips; Echelon Biosciences) were blocked by incubation for 1 h with 1% fatty-acid free BSA in TBST (137 mm NaCl, 2.37 mm KCl, 19 mM Tris base, 1% Tween 20). All incubations were carried out at room temperature. We incubated 10 ml of PBST supplemented with 1% fatty-acid free BSA and 10 μg of either of the purified N-terminal cytosolic domains, all of which were GFP-tagged, for 1 h with the PIP strips. The PIP strips were washed four times with TBST and incubated for 1 h with a 1:1000 dilution of anti-GFP antibodies in TBST supplemented with 1% fatty-acid free BSA. The membrane was again washed four times and incubated with a 1:10,000 dilution of horseradish peroxidase-conjugated anti-mouse antibodies. Bound antibodies were detected by chemiluminescence with the SuperSignal West Pico Chemiluminescent Substrate (ThermoScientific, #34080) and the C-Digit Blot Scanner Imaging System (LiCor).

**Split-GFP assay**. The split GFP system is based on GFP fragments containing β-strands 1–10 (GFP₁–₁₀) and β-strand 11 of GFP (GFP₁₁) reconstituting complete β-barrel structure of GFP able to emit fluorescence when in sufficient proximity[92–94]. COS-7 cells were transfected either with single constructs to verify their expression, localization and absence of signal in GFP channel, or double transfected to evaluate for the presence of the signal in the GFP channel. The imaging was done on Leica SP8 confocal microscope with same imaging parameters between conditions. Since TMEM16K forms a dimer[33,56], we validated the split-GFP approach by expressing TMEM16K-GFP₁–₁₀ and GFP₁₁-TMEM16K to reconstitute the split-GFP. We used proteins that are not considered TMEM16K interaction partners based on our proteomics, but are known to participate in similar processes/compartments (OSBPL8, OSBPL9, VPS35) as negative controls. Images are represented using pseudocolors suitable for color-blind palette.

**Statistical analysis**. We used one-tailed t-test, two-tailed t-test or single factor ANOVA with post-test Bonferroni-corrected two sided t-test. We used box plot to graphically visualize data with all box-plot elements defined in the same manner for every box-plot used in this manuscript; the box includes the first quartile and the third quartile, with the central line representing the median. Whiskers represent the minimum and maximum values of data. X inside the box represents the mean of data. No statistical method was used to determine sample size in any of the experiments.

**Reporting summary**. Further information on research design is available in the Nature Research Reporting Summary linked to this article.

## Data availability
The data that support the findings of this study are available from the corresponding author upon reasonable request. The source data underlying Figs. 1b, 1c, e, g, 3a, b, 4a–f, 4h, 6g, 7d, 8f, Supplementary Figs 3b, c, Supplementary Figs. 5b, 5e are provided as a Source Data file. Our proteomics datasets are available via ProteomeXchange with identifier PXD018990. We have used publicly available String and SwissProt databases.

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

## Acknowledgements

We thank Matthew Klassen, Vanja Stojkovic, Caitlin O'Brien, David Crottes, Beverly Pigott and Damien Jullie for critical reading of the manuscript, and members of the Jan laboratory for discussion. We acknowledge USCF Nikon Imaging Center, Larsen Delaine and Kari Herrington for imaging support. The electron microscopy was performed by Ivy Hsien at the San Francisco Veteran Affairs Medical Center (VAMC), for which we are very thankful. This work was supported by National Institutes of Health grant (R35NS097227 and R21AG061468) to YNJ and National Institutes of Mental Health grant (R37MH0653354) to LYJ. MP was supported by a Fyssen Postdoctoral Fellowship and National Ataxia Young Investigator Award. Mass spectrometry analysis was provided by the Bio-Organic Biomedical Mass Spectrometry Resource at UCSF (A.L. Burlingame, Director) supported by the Biomedical Technology Research Centers program of the NIH National Institute of General Medical Sciences, NIH NIGMS 8P41GM103481 and NIH 1S10OD016229). YNJ and LYJ are investigators of the Howard Hughes Medical Institute.

## Author contributions

MP conceived of the project. MP designed and performed cellular and mouse experiments. MP, AB, and JOP designed, and AB and JOP performed the mass spectrometry analysis. MP, YNJ, and LYJ wrote the manuscript.

## Competing interests

The authors declare no competing interests.
