## [Peer Review File · Nature Communications]

Reviewers' comments:

Reviewer #1 (Remarks to the Author):

This paper from the Jan lab provides evidence that TMEM16K plays a role in endosomal transport and suggests that mutations in TMEM16K may produce SCAR10 as a result of defects in endosomal sorting. This is an extremely interesting, but sprawling, investigation that opens new territory in both the TMEM16 and membrane trafficking fields. While I am highly intrigued by its novelty, overall I think that the conclusions are drawn in a somewhat cavalier manner from data that is preliminary or incomplete.

Major issues:

There is no characterization of the knockout mouse to show that it does not express TMEM16K.

The authors begin by showing that the TMEM16K knockout mice exhibit hindlimb clasping and smaller neuromuscular junctions, but the paper does not provide an explanation. While the implication that this phenotype can be explained by defective endosomal trafficking is appealing, there are no data to allow such a conclusion.

The proteomic analysis in Fig. 2 and S-Table 1 is not well presented, confusing, and appears misleading. S-Table 1 shows <50 proteins, but Fig. 2 shows 365 proteins. The correspondence between the table and figure is inscrutable and it is not clear how much of Fig. 2 is actual data and how much is bioinformatics. For example, of the 16 proteins in S-Table 1 listed as Endosomal transport, only 5 appear in the pink panel labelled Endosomal Transport in Fig. 2c. The majority of the proteins in this panel do not appear in S-Table 1.

Fig. 3a,b shows nicely that the trafficking of CI-MPR and CtxB are altered in the knockout, but does this mean that endosome-to-Golgi trafficking is reduced, or that retention in the Golgi is reduced possibly by increased anterograde traffic? Experiments to sort this out are necessary to justify the conclusion that "TMEM16K is required for proper endosome to trans-Golgi retrograde trafficking".

The localization of TMEM16K in ER is not quantitatively established. Although the data show that the bulk of it is in ER, expression in the endolysosomal system at a lower level is not investigated rigorously. The authors' interpretation of the split GFP experiments assumes that 16K is exclusively localized in ER and Rab7 exclusively in endosomes. If even a small fraction of 16K and Rab7 were localized in the same compartment, one might expect to see reconstitution of fluorescence.

The authors do not show why lysosomal pH is altered. Does it alter the trafficking of lysosomal channels that regulate pH (like the v-H-ATPase)? If so, this should be tested. If 16K is present in lysosomes and, like other TMEM16s, conducts ions, it might alter lysosomal pH directly. More information is required to allow the conclusion that (Page 4) "these results show that loss of TMEM16K causes a defect in endosomal retrograde sorting, which can cause deficiencies within the later stages of the endolysosomal system".

I do not know how to interpret Fig. 4a and b. The bright SA labeling in Fig. 4a seems to enclose a large cluster of Rab7 positive structures (~2 um in diameter) and colocalization of 16K and Rab7 seems minimal. But in Fig. 4b, 16K and Rab7 co-localize perfectly, which is not what one would expect if ER and endosomes are interacting. A more quantitative analysis is required.

The conclusion that "ER-localized TMEM16K forms contact sites with endosomes, interacting with Rab7 and PtdIns(3)P" is not justified. The authors only show that PI3P binds to 16K N-terminus in an in vitro system and that knockout of TMEM16K alters the size of PI3P-positive vesicles.

Specific comments.

Because there is some variability in the literature, the authors should state exactly what the box plots mean.

Page 2. "While the yeast homolog Ist2 forms MCS that play a vital role in lipid homeostasis at MCS between the ER and plasma membrane. " IST2 has not been shown to participate in lipid homeostasis, to my knowledge.

Page 2. "mammalian family member most closely related to Ist2." I think that this is an exaggeration. The mammalian homologs have only 5 – 9% identity to Ist2. Although TMEM16K may have the highest percent identity (9%), any implication that TMEM16K has any functional similarity to Ist2 is unwarranted. Fig. 1a should have a scale bar and bootstrap values.

S-Fig.1c and S-Fig. 2a,b are interpreted too casually. While the colocalization of 16K with ER-tdTomato is convincing, the co-localization with calreticulin and PDI is less obvious. It seems that PDI and 16K are in different sub-compartments even though there is general overlap. A quantitative treatment of these data is necessary. Also, because overexpressed proteins often accumulate in the ER, is there any data that endogenous TMEM16K is concentrated in ER?

Page 2. "knockout mice displayed increasing hindlimb clasping, a behavioral phenotype characteristic of impaired neuromuscular function." Please provide a reference. It seems to me that hindlimb clasping is more likely explained by CNS defects rather than defective neuromuscular junctions.

Page 3. "Because dysfunctions of endosomal transport are tightly associated with neurodegenerative diseases." What does "tightly associated with" mean?

Fig. 3f legend: (WT slope= -0.633, KO slope= -0.0438). These numbers are in error. There is not a >10-fold difference in slopes. Also, please provide units. There are no error bars.

Fig. 3g. It is not clear how these experiments were performed: is this a steady-state measurement in the absence of FCCP? Because Lysosensor is not ratiometric is it ratioed vs. Cherry-CAAX fluorescence? Are these measurements reliable without a ratiometric indicator?

Fig. 4d – Is the box in 16K-V5-GFP11 drawn correctly to show enlarged area?

Images showing overlay of 3 colors are not useful because there is too much information. I would prefer to see overlays of 2 key channels at a time so that I can clearly see colocalization.

Reviewer #2 (Remarks to the Author):

There is great interest in the composition and function of membrane contact sites, and this paper has used BioID to identify proteins in the vicinity of TMEM16K, an ER protein that is a lipid scramblase linked to spinocerebellar ataxia. In mouse ubiquitous and neuron specific knock outs, the authors observed reductions in the sizes of neuromuscular junctions and overall impaired neuromuscular function. This was not well connected to the rest of the story. Upon BioID, the authors detected ER and endosomal protein interactions. The authors noted dispersal of mannose 6-phosphate receptors from a perinuclear compartment and decreased transport of cholera toxin to the Golgi as well as weaker endo-lysosomal acidification which could have been due to defects in trafficking of newly synthesized lysosomal enzymes and the V-ATPase. (Note that MPRs usually are seen in perinuclear late endosomes, not the TGN at steady state). The authors note an

interaction with Rab7 by split GFP and STED reveals compartment interaction. (A nice control would have been a GDP-locked Rab7). Dot blots suggest that the protein binds PI3P but Rab7 is late endosomal where there would be more PI3,5P. Because dot blots aren't perfect, it might be worth trying to recapitulate this interaction in a liposome flotation scenario, although the accumulation of PI3P structures is consistent with what was shown. Interestingly, TMEM16K seems to influence PI3P levels independent of VAPA/B, enlarging these structures. Is it blocking early to late endosome Rab conversion? Transferrin recycling was not altered, nor was EGF co-localization with Rab7. Longer half time of EGF degradation would provide indication of whether conversion was altered and this is easy to add and should be.

In summary, there is a lot of nice data here implicating a lipid scramblase in a new class of junction. The precise mechanism is not yet shown but the data will be of interest to workers in this area. Perhaps the authors could look at the consequences of KNOCKOUT on PikFYVE kinase that generates PI3,5P as its activity seems to be lacking. Also, the authors should try to link the mouse phenotypes with the cell biology.

Reviewer #3 (Remarks to the Author):

The manuscript of Petkovic et al. focuses on the role of the lipid scramblase TMEM16K as a potential regulator of endosomal sorting. The authors demonstrate that TMEM16K contact proteins of the endosomal machinery, likely via contact sites, and interacts with Rab7 and PI-3-P. Loss of TMEM16K impairs endosomal transport, yet not other trafficking pathways. The authors suggest that TMEM16K is a critical factor in ER-endosome contact site formation and endosomal sorting.

The authors provide a full spectrum of assays to address the critical role of TMEM16K in endosomal biogenesis. Their analysis of the knock-out mutant is overall extensive, and complementation analyses suggest that the scramblase activity is required for TMEM16K function. While I consider the extent of their analysis impressive, I miss several controls to be able to judge the relevance of their observations relative to the loss of known endosomal proteins. Also lipid scramblase activity needs are not really addressed. My detailed points are listed below:

1. Figure 3 is the first figure where defects in TMEM16K are addressed. The authors need to show a positive control for endosomal transport defects (a,b). Otherwise it is not possible to judge the relevance. Is this a strong defect or a mild one? The same applies for Figure 3e-g. There is no knock out analysis shown without complementation with wild-type or mutant 16K, or any other known mutant/siRNA knock out affected in lysosomal biogenesis. This is again critical to judge if TMEM16K is just needed as a supporting factor or has an overall critical role.
2. Figure 3b does not show complementation with the wild-type protein. This needs to be added. Also Figure 3e is incomplete here.
3. Along the same line, the figure has to be completed with information provided in Suppl. Figure 1 regarding the ER localization. Here as along the entire manuscript, the content of the figures is so condensed that the paper becomes unreadable without having the supplements next to each figure.
4. Figure 4a is impossible to read. There is not cell outline and I am just completely puzzled what to look at.
5. Figure 4g suggests that TMEM16K has a PI-3-P binding domain. This needs to be repeated with recombinant proteins and a flotation assay. I agree that this is indicative, yet it remains incomplete. In addition, the identified PIP-binding deficient mutant needs to be analyzed in their complementation (Figure 3a, b, e) and interaction analyses (Figure 4d).
6. The authors could here also use a complementation assay by mutating the PIP-binding domain and by fusing TMEM16K to a PI-3-P interacting domain. If their interpretation is correct, this should rescue the mutant (if it indeed has a defect in endosomal sorting).
7. Figure 5: The authors suggest that scramblase activity of TMEM16K is important for endosomal transport. However, their mutants in Figure 5 are not analyzed for deficient scramblase activity.

This is particularly important for the parts where entire segments are swapped.

8. I normally would expect a detailed ultrastructural analysis of endosome-ER contact sites and their alterations due to TMEM16K mutants. Has this been done?

Response to reviewers' comments

We thank the reviewers for their helpful and constructive comments. We have significantly improved our paper by performing several additional experiments and modified the figures and the text of the manuscript accordingly. With the new data, we have confirmed as well as extended our conclusions from the original submission. We believe that we have addressed the reviewers' comments whenever possible and have substantially improved the paper. We hope that the reviewers will now find our revised paper suitable for *Nature Communication*.

The following is a brief summary of the major changes made and is followed by our point-by-point response to referees' comments.

- We have done colocalization analysis of TMEM16K with ER markers Calreticulin, PDI and Sec61 β . We have also included colocalization analysis of Sec61 β with other ER-markers to provide meaningful context for TMEM16K colocalization analysis, as Sec61 β is a pore forming subunit of translocon complex localized exclusively to ER (Fig. 1e,f)
- We have evaluated whether reintroducing TMEM16K into TMEM16K knockout cells rescues the endosomal retrograde trafficking defect observed with cholera toxin B (CtxB) by (Fig. 3b).
- We have evaluated anterograde biosynthetic pathway using "Retention Using Selective Hooks" (RUSH) system. RUSH allows synchronization of protein transport through the biosynthetic pathway (Boncompain et al., 2012) and we tracked three transmembrane proteins with different steady state distributions: the glycosylphosphatidylinositol anchor (GPI; transported to plasma membrane), the transferrin receptor (TfR; transported to plasma membrane, early endosomes, and recycling endosomes), and the cation-dependent mannose-6-phosphate receptor (CD-MPR; transported from TGN directly to early/late endosomes). (Fig. 3c-e).
- We have performed EGF colocalization experiment with Rab7 with longer EGF incubation time points (40 min and 60 min). (Fig. 3g)
- We have redone the imaging experiments evaluating TMEM16K proximity to endosomes to include the view of the entire cells, region of interest as well as high magnification incepts. We have also included line scan quantifications. (Fig. 4a-e)
- We have included live imaging of TMEM16K with endosomes labeled with Rab7 and EGF. (Fig. 4b, Supplemental Video 3)
- To further evaluate the specificity of TMEM16K interaction with the Rab7, we generated Rab7 mutants: constitutively active Q67L, which mimic permanently GTP-bound Rab7 and inactive T22N, which mimic permanently GDP-bound Rab7. We have tested their ability to reconstitute split-GFP with TMEM16K (Fig. 4e,j).
- We have generated N-terminal truncation of TMEM16K, and evaluated its ability to reconstitute split-GFP with Rab7, as well as its ability to rescue endosomal retrograde trafficking defect (Fig. 5d, e,f).
- We have tested effects of pharmacological inhibition of PikFYVE kinase in the wild type and TMEM16K knockout cells (Fig. 6c,d)
- We have evaluated human disease point mutants and scrambling domains chimeras for their ability to reconstitute split-GFP with Rab7 (Fig. 6g,h)
- We have done RT-PCR analysis evaluating presence of TMEM16K mRNA in liver and brain tissues obtained from the wild type and TMEM16K ubiquitous knockout littermates. (Supplemental Fig. 1b)

Point by point response to Reviewer Comments:

Reviewer #1 (Remarks to the Author):

This paper from the Jan lab provides evidence that TMEM16K plays a role in endosomal transport and suggests that mutations in TMEM16K may produce SCAR10 as a result of defects in endosomal sorting. This is an extremely interesting, but sprawling, investigation that opens new territory in both the TMEM16 and membrane trafficking fields. While I am highly intrigued by its novelty, overall I think that the conclusions are drawn in a somewhat cavalier manner from data that is preliminary or incomplete.

Major issues:

There is no characterization of the knockout mouse to show that it does not express TMEM16K.

We have added RT-PCR results showing the absence of TMEM16K mRNA in TMEM16K knockout mice (Supplemental Fig. 1b). Information about all the primers for RT-PCR as well as genotyping are in the Material and Methods, along with the identification information from the International Knockout Mouse Consortium that generated the TMEM16K conditional knockout mice used in our study.

The authors begin by showing that the TMEM16K knockout mice exhibit hindlimb clasping and smaller neuromuscular junctions, but the paper does not provide an explanation. While the implication that this phenotype can be explained by defective endosomal trafficking is appealing, there are no data to allow such a conclusion.

While we would very much like to provide direct demonstration that the observed defects in endosomal transport are the cause of the pathology, achieving this goal would require demonstration that the observed cellular defects can be rescued in a way different from reintroducing TMEM16K to mutant cells (for example with small molecule) and the same treatment can also relieve the pathology. While no such treatment has been identified, we have tested the converse scenario, and shown that mutations of three different single amino acids in TMEM16K that cause the pathology in humans also severely impact the ability of these TMEM16K mutants to rescue the observed cellular defects (Fig. 6g,i,j).

The proteomic analysis in Fig. 2 and S-Table 1 is not well presented, confusing, and appears misleading. S-Table 1 shows <50 proteins, but Fig. 2 shows 365 proteins. The correspondence between the table and figure is inscrutable and it is not clear how much of Fig. 2 is actual data and how much is bioinformatics. For example, of the 16 proteins in S-Table 1 listed as Endosomal transport, only 5 appear in the pink panel labelled Endosomal Transport in Fig. 2c. The majority of the proteins in this panel do not appear in S-Table 1.

Thank you for helpful comment. We have clarified the text.

Briefly, Fig. 2b shows actual raw data -the obtained candidates from the TMEM16K proteomics. To identify in a unbiased manner the most biologically relevant functional clusters which could infer TMEM16K function rather than just hand-picking one or two highly ranked candidates for our study, we have visualized the list of these candidates as protein-protein interaction network using String app in the Cytoscape.

From here, we have done 2 bioinformatics analysis:

(1) We identified functional enrichment categories based on GO terms, which we overlaid with color-code on our list of candidates. Indeed, visualization of this overlay suggest functional clusters of proteins identified in the TMEM16Kproteomics.

(2) We performed a mathematical clustering analysis using MCODE cluster app in Cytoscape to identify clusters from our raw data, generating the simplified network shown in Fig. 2c. Clusters are defined as highly interconnected nodes in the network. We then overlaid the color-coding from functional enrichment analysis on this simplified network generated via clustering analysis.

We have originally listed only a subset of the proteins listed in the Supplemental Table 1 to indicate their cellular functions. Thank you for your helpful comment to point out that this is confusing - we have added the full list of candidates to Supplemental Table 1. In addition, all the raw proteomics data will be made available to other researchers.

Fig. 3a,b shows nicely that the trafficking of CI-MPR and CtxB are altered in the knockout, but does this mean that endosome-to-Golgi trafficking is reduced, or that retention in the Golgi is reduced possibly by increased anterograde traffic? Experiments to sort this out are necessary to justify the conclusion that “TMEM16K is required for proper endosome to trans-Golgi retrograde trafficking”.

CI-MPR and CTxB are classical markers used to evaluate endosome to TGN retrograde trafficking by performing pulse chase experiments, so in these experiments we are following only the retrogradely trafficked molecules from the plasma membrane through endosomes to TGN. Their mis-localization in pulse chase experiments shows that the endosome to trans-Golgi retrograde trafficking is perturbed as previously reported in literature (Bonifacino and Rojas, 2006; Sandvig and van Deurs, 2002; Seaman et al., 1998). However, as reviewer pointed out, one can imagine a scenario in which anterograde secretory pathway is perturbed upstream of retrograde trafficking from endosomes to TGN. To address this question, we took advantage of recent methodological developments that allow synchronization of protein transport through the biosynthetic pathway (Chen et al., 2017). Using the “Retention Using Selective Hooks” (RUSH) system we tracked the biosynthetic transport of three transmembrane proteins with different steady state distributions: the glycosylphosphatidylinositol anchor (GPI; transported to plasma membrane), the transferrin receptor (TfR; transported to plasma membrane, early endosomes, and recycling endosomes), and the cation-dependent mannose-6-phosphate receptor (CD-MPR; transported from TGN directly to early/late endosomes). We found no difference in the transport through the biosynthetic pathway between the MEF cells from wild type mice and TMEM16K knockout mice, showing that the anterograde secretory pathway is unaffected (Fig. 3c,d,e).

The localization of TMEM16K in ER is not quantitatively established. Although the data show that the bulk of it is in ER, expression in the endolysosomal system at a lower level is not investigated rigorously.

We have shown ER localization in both live imaging and immunofluorescence (Fig. 1e, Supplemental Video 2). We have quantified the colocalization of TMEM16K with Calreticulin, PDI and Sec61 β (Fig. 1f). We have also done immunohistochemistry and quantified in the same manner the colocalization of Sec61 β with Calreticulin and PDI to provide a meaningful context for the colocalization analysis obtained with TMEM16K (Fig. 1f), given that Sec61 β is a pore forming component of the translocon complex localized exclusively to the ER.

Additionally, Bushnell et al. reported in *Nature Communication* (2019) TMEM16K crystal structure as well as an analysis of the TMEM16K localization to the ER, further corroborating our findings.

The authors' interpretation of the split GFP experiments assumes that 16K is exclusively localized in ER and Rab7 exclusively in endosomes. If even a small fraction of 16K and Rab7 were localized in the same compartment, one might expect to see reconstitution of fluorescence.

Thank you for raising this point. We have tested several other endosomal proteins found in our proteomics (OSBPL11, SNX1, SNX2, VPS26) with the split-GFP assay (Supplemental Fig. 4) and did not observe positive GFP signal with any of the other tested proteins besides Rab7 (Fig. 4h), so it seems unlikely that a small fraction of the TMEM16K is in the endosomal compartment. We have also found that TMEM16K mutants (F337V, D615N, SCRDA), which are all localized in the ER, are not able to reconstitute the split-GFP with Rab7 (Fig. 6g,h), further supporting the specificity of this interaction. Likewise, TMEM16K was able to reconstitute split-GFP only with the constitutively active Rab7 mutant Q67L (Fig. 4i), but not with the inactive T22N Rab7 mutant (Fig. 4j), again corroborating the specificity of the interaction.

The authors do not show why lysosomal pH is altered. Does it alter the trafficking of lysosomal channels that regulate pH (like the v-H-ATPase)? If so, this should be tested. If 16K is present in lysosomes and, like other TMEM16s, conducts ions, it might alter lysosomal pH directly. More information is required to allow the conclusion that (Page 4) "these results show that loss of TMEM16K causes a defect in endosomal retrograde sorting, which can cause deficiencies within the later stages of the endolysosomal system".

As pointed out by the reviewer, V-ATPase and other lysosomal channels indeed could have perturbed trafficking, and this will be the focus of follow-up studies. However, it does not seem likely TMEM16K is directly contributing to altering lysosomal pH—TMEM16K was shown to be a scramblase and we and others have localized it to the endoplasmic reticulum (Bushnell et al., 2019). We have revised the statement.

I do not know how to interpret Fig. 4a and b. The bright SA labeling in Fig. 4a seems to enclose a large cluster of Rab7 positive structures (~2 um in diameter) and colocalization of 16K and Rab7 seems minimal. But in Fig. 4b, 16K and Rab7 co-localize perfectly, which is not what one would expect if ER and endosomes are interacting. A more quantitative analysis is required.

Thank you for these helpful comments. To address these comments, we have included live imaging showing fluorescently tagged TMEM16K, with fluorescently tagged Rab7 and EGF-Alexa647 (Fig. 4b, Supplemental Video 3). These experiments reveal highly dynamic movements of TMEM16K and Rab7, as well as their interactions. We have also added line scans for a quantitative analysis.

The conclusion that "ER-localized TMEM16K forms contact sites with endosomes, interacting with Rab7 and PtdIns(3)P" is not justified. The authors only show that PI3P binds to 16K N-terminus in an in vitro system and that knockout of TMEM16K alters the size of PI3P-positive vesicles.

We have used a broad range of approaches to support our conclusion that TMEM16K forms contacts with Rab7-endosomes, including proximity biotinylation proteomics (Fig. 2), confocal/live imaging/super-resolution microscopy (Fig. 4), and split-GFP

reconstitution (Fig. 4, 5, 6), by following precedence in the literature (Scorrano et al., 2019). We have validated the specificity of this interaction using mutants of both Rab7 and TMEM16K (Fig. 4i,j, 6g,h). We have further shown *in vitro* binding of a subdomain of TMEM16K to a subset of phosphatidylinositols, with two other mammalian members used as control (Fig. 5a-c), as done for various phosphatidylinositol binding proteins (Fischer et al., 2009; Kim et al., 2008). We have also found that the absence of TMEM16K leads to enlarged PI3P-vesicles, and showed that reintroducing TMEM16K can rescue the defect (Fig. 6b,d), as it was similarly done for Ist2 and PtdIns4P at the plasma membrane (Fischer et al., 2009; Manford et al., 2012). We have modified the wording to more precisely represent our findings: “TMEM16K forms contact sites with endosomes, reconstituting split-GFP with small GTPase Rab7 and binding phosphatidylinositol 3-phosphate (PtdIns(3)P).”

Specific comments.

Because there is some variability in the literature, the authors should state exactly what the box plots mean.

We have added the following clarification to the Statistics section in Materials and Methods: “We used box plot to graphically visualize data where the box includes the first quartile and the third quartile, with the central line representing the median. Whiskers represent the minimum and maximum values of data. X inside the box represents the mean of data.”

Page 2. “While the yeast homolog Ist2 forms MCS that play a vital role in lipid homeostasis at MCS between the ER and plasma membrane. “ IST2 has not been shown to participate in lipid homeostasis, to my knowledge.

Manford et al. *Developmental Cell* (2012) reported the accumulation of plasma membrane PtdIns4P in the absence of Ist2. Such accumulation leads to perturbed exchange of lipids at the ER-plasma membrane MCS. We have included this citation in the revised manuscript.

Page 2. “mammalian family member most closely related to Ist2.” I think that this is an exaggeration. The mammalian homologs have only 5 – 9% identity to Ist2. Although TMEM16K may have the highest percent identity (9%), any implication that TMEM16K has any functional similarity to Ist2 is unwarranted. Fig. 1a should have a scale bar and bootstrap values.

While 9% is low identity, this was what sparked our interest to pursue the question whether TMEM16K might be involved in membrane contact sites. With a similarly low percentage of amino acid identity between TMEM16 family members, there is functional similarity among lipid scramblases such as mammalian TMEM16K (Bushell et al., 2019), mammalian TMEM16F (Yang et al., 2012), fungal afTMEM16 (Malvezzi et al., 2013) and nhTMEM16 (Brunner et al., 2014), as well as a distant ameboidea DfTMEM homolog (Pelz et al., 2018). Considering that the mammalian TMEM16 family members are modulators of diverse cellular functions, it will be of significant interest and physiological relevance to determine whether some members of this family have a role in interorganelle communication. As we now state in our revised manuscript, since submission of this manuscript, another study was recently published showing that mammalian TMEM16H

has a role at membrane contact sites between ER and plasma membrane (Jha et al., 2019).

S-Fig.1c and S-Fig. 2a,b are interpreted too casually. While the colocalization of 16K with ER-tdTomato is convincing, the co-localization with calreticulin and PDI is less obvious. It seems that PDI and 16K are in different sub-compartments even though there is general overlap. A quantitative treatment of these data is necessary. Also, because overexpressed proteins often accumulate in the ER, is there any data that endogenous TMEM16K is concentrated in ER?

Localization of the endogenous proteins compared to tagged version can be an issue. We have tested commercially available antibodies and generated 2 different TMEM16K antibodies to visualize endogenous protein, but we found the antibodies were not specific. Hence, we have used a multitude of different tags and have found that all the constructs yielded only ER localization as assessed by both live imaging and immunofluorescence (Fig. 1e,f, Supplemental Video 2). We have performed quantification of colocalization between TMEM16K and ER markers Calreticulin, PDI and Sec61 β . We further cite in our revised manuscript the recent publication of Bushell et al. in *Nature Communications* (2019), which reports TMEM16K crystal structure as well as TMEM16K localization to endoplasmic reticulum.

Page 2. “knockout mice displayed increasing hindlimb clasping, a behavioral phenotype characteristic of impaired neuromuscular function.” Please provide a reference. It seems to me that hindlimb clasping is more likely explained by CNS defects rather than defective neuromuscular junctions.

Hindlimb clasping is a behavioral marker of disease progression in a number of mouse models of neurodegeneration, including ataxias, and is characteristic of impaired neuromuscular function (Guyenet et al., 2010; Hatzipetros et al., 2015). As reviewer pointed out, this impaired neuromuscular function indeed can be due to defects in the CNS including the spinal cord (Lalonde and Strazielle, 2011). We also need to bear in mind that neuromuscular junctions correspond to the synaptic contacts between muscle and axon terminals of motor neurons in the CNS. Reduction of neuromuscular junctions is an early pathological target of impaired neuromuscular function that is easily accessible for histological visualization and quantification (Sleigh et al., 2014). We have included citations in the revised manuscript.

Page 3. “Because dysfunctions of endosomal transport are tightly associated with neurodegenerative diseases.” What does “tightly associated with” mean?

With an increasing number of genes involved in the endolysosomal pathway implicated in neurodegenerative diseases, endolysosomal dysfunction represents a pathophysiological mechanism shared across these diseases (Neefjes and van der Kant, 2014; Wang et al., 2018). We have provided additional references in the revised manuscript.

Fig. 3f legend: (WT slope= -0.633, KO slope= -0.0438). These numbers are in error. There is not a >10-fold difference in -slopes. Also, please provide units. There are no error bars.

Thank you, indeed this is an error. We have corrected the mistake and provided the equations and R².

Fig. 3g. It is not clear how these experiments were performed: is this a steady-state measurement in the absence of FCCP? Because Lysosensor is not ratiometric is it ratioed vs. Cherry-CAAX fluorescence? Are these measurements reliable without a ratiometric indicator?

Thank you for a well raised point. Lysosensor green DNP-189 is extensively used in literature as a reliable way of qualitatively evaluating pH of the acidic compartments (Davis-Kaplan et al., 2004; Lu et al., 2014; McKnight et al., 2014; Pi et al., 2017). We have performed the experiments in the following manner: we loaded Lysosensor Green DNP-189 over 30 min incubation, washed twice with imaging media and all acquisition was done over the next 3 min, where we measured the fluorescence intensity per cell.

We have additionally tested the specificity of the measured signal with Lysosensor Green DNP-189 using proton ionophore FCCP (Fig. 3i) Furthermore, we have performed extensive rescue experiment with both wild type and mutant TMEM16K cDNA (Fig. 3j).

Fig. 4d – Is the box in 16K-V5-GFP11 drawn correctly to show enlarged area?

Thank you, indeed the box marking the inset on the 16K-V5-GFP11 image was shifted. We have corrected it.

Images showing overlay of 3 colors are not useful because there is too much information. I would prefer to see overlays of 2 key channels at a time so that I can clearly see colocalization

We were unsure which two channels would constitute key channels, so we have added the line scan quantification to simplify interpretation of the colocalization (Fig 4a,b,e). We hope this helps in evaluating the represented data.

Reviewer #2 (Remarks to the Author):

There is great interest in the composition and function of membrane contact sites, and this paper has used BioID to identify proteins in the vicinity of TMEM16K, an ER protein that is a lipid scramblase linked to spinocerebellar ataxia. In mouse ubiquitous and neuron specific knock outs, the authors observed reductions in the sizes of neuromuscular junctions and overall impaired neuromuscular function. This was not well connected to the rest of the story.

As mentioned previously, while we would very much like to provide direct demonstration that the observed defects in endosomal transport are the cause of the pathology, this would require that we can rescue the observed cellular defects in a different way (for example with small molecule) so that we can test whether this treatment that rescues the cellular defects also relieve the pathology. Before such a treatment can be developed, we have tested the converse scenario, and shown that three different TMEM16K mutations with single amino acid substitution that cause the pathology in humans also compromise the ability of TMEM16K to rescue the observed cellular defects.

Upon BioID, the authors detected ER and endosomal protein interactions. The authors noted dispersal of mannose 6-phosphate receptors from a perinuclear compartment and decreased transport of cholera toxin to the Golgi as well as weaker endo-lysosomal acidification which could have been due to defects in trafficking of newly synthesized lysosomal enzymes and the V-ATPase. (Note that MPRs usually are seen in perinuclear late endosomes, not the TGN at steady state).

To avoid the potential caveats mentioned by the reviewer, we pursue pulse chase experiments looking at trafficking of internalized CI-MPR from the plasma membrane through endosome to TGN. We have clarified the text that we are looking at the pulse chase experiment at the t=60min time point, and not a steady state localization.

The authors note an interaction with Rab7 by split GFP and STED reveals compartment interaction. (A nice control would have been a GDP-locked Rab7).

Thank you for the great suggestion. We have generated inactive GFP(1-10)-HA-T22N Rab7 and constitutively active GFP(1-10)-HA-Q67L Rab7 mutants (Spinosa et al., 2008), and evaluated their ability to reconstitute GFP with TMEM16K-V5-GFP11. We found that TMEM16K was able to reconstitute split-GFP only with the constitutively active Rab7 Q67L mutant (Fig. 4i), but not with the inactive Rab7 T22N mutant (Fig. 4j), further validating the specificity of the observed contact between TMEM16K and Rab7 endosomes.

Dot blots suggest that the protein binds PI3P but Rab7 is late endosomal where there would be more PI3,5P. Because dot blots aren't perfect, it might be worth trying to recapitulate this interaction in a liposome flotation scenario, although the accumulation of PI3P structures is consistent with what was shown. Interestingly, TMEM16K seems to influence PI3P levels independent of VAPA/B, enlarging these structures. Is it blocking early to late endosome Rab conversion? Transferrin recycling was not altered, nor was EGF co-localization with Rab7. Longer half time of EGF degradation would provide indication of whether conversion was altered and this is easy to add and should be.

Thank you for this great suggestion. We have performed this experiment looking at the 40min and 60min time point. We found no difference between the wild type and knockout cells in the colocalization of the EGF with the late endosomal markers Rab7 at 10 min, 15 min and 40 min time points (Fig. 3g), indicating that the mutant phenotype arose from a defect at or after the Rab7 stage of endolysosomal maturation. However, we found greater retention of the EGF in Rab7 endosomes at 60 min in TMEM16K knockout cells compared to wild type cells (Fig. 3g), suggesting defect in endosomal sorting.

In summary, there is a lot of nice data here implicating a lipid scramblase in a new class of junction. The precise mechanism is not yet shown but the data will be of interest to workers in this area. Perhaps the authors could look at the consequences of KNOCKOUT on PikFYVE kinase that generates PI3,5P as its activity seems to be lacking

Thank you for another great suggestion. We have tried evaluating PtdIns(3,5)P₂ with the available lipid sensor ML1Nx2, but we were not convinced it could be reliably used for studies of primary mouse fibroblasts. Hence we used pharmacological manipulation with YM201636 to inhibit PikFYVE kinase to evaluate the observed phenotypes in the wild type and TMEM16K knockout cells using PtdIns3P lipid sensor P40XP-gfp. Inhibiting PIKfyve in TMEM16K wild type cells recapitulated the TMEM16K knockout phenotype. However, in the TMEM16K knockout cells we did not observe additional cumulative effect, suggesting that conversion between PtdIns3P to PtdIns(3,5)P₂ is impaired in the absence of TMEM16K (Fig. 6c,d).

Also, the authors should try to link the mouse phenotypes with the cell biology.

We wholeheartedly agree with the reviewer that it would be great to further investigate how the observed cellular defects lead to the mouse phenotypes. Such an endeavor will require developing new ways to rescue the cellular defects so as to test whether rescue approaches other than reintroducing wildtype TMEM16K, but not TMEM16K bearing disease-causing mutations, can also rescue the mouse phenotypes – a fairly substantial new project. We feel that sharing our findings with the community at this point before embarking another demanding project will be of interest; our revised manuscript will contribute to the better understanding of this link by facilitating the development of follow up studies by our lab as well as our colleagues.

Reviewer #3 (Remarks to the Author):

The manuscript of Petkovic et al. focuses on the role of the lipid scramblase TMEM16K as a potential regulator of endosomal sorting. The authors demonstrate that TMEM16K contact proteins of the endosomal machinery, likely via contact sites, and interacts with Rab7 and PI-3-P. Loss of TMEM16K impairs endosomal transport, yet not other trafficking pathways. The authors suggest that TMEM16K is a critical factor in ER-endosome contact site formation and endosomal sorting.

The authors provide a full spectrum of assays to address the critical role of TMEM16K in endosomal biogenesis. Their analysis of the knock-out mutant is overall extensive, and complementation analyses suggest that the scramblase activity is required for TMEM16K function. While I consider the extent of their analysis impressive, I miss several controls to be able to judge the relevance of their observations relative to the loss of known endosomal proteins. Also lipid scramblase activity needs are not really addressed. My detailed points are listed below:

1. Figure 3 is the first figure where defects in TMEM16K are addressed. The authors need to show a positive control for endosomal transport defects (a,b). Otherwise it is not possible to judge the relevance. Is this a strong defect or a mild one? The same applies for Figure 3e-g. There is no knock out analysis shown without complementation with wild-type or mutant 16K, or any other known mutant/siRNA knock out affected in lysosomal biogenesis. This is again critical to judge if TMEM16K is just needed as a supporting factor or has an overall critical role.

We are grateful for your correction. Indeed, the point of our experiments was not to claim that TMEM16K is more or less important than other proteins shown to regulate endosomal retrograde trafficking, but to show that it is involved in this process, and that in absence of this ER protein endosomal retrograde trafficking is perturbed. We have corrected the text appropriately to avoid misconception. Many labs have done important work studying this pathway, and we feel it would be outside of the scope of this paper to expand it to include hierarchical analysis with other known regulators.

2. Figure 3b does not show complementation with the wild-type protein. This needs to be added. Also Figure 3e is incomplete here.

Thank you for pointing it out. We have added the rescue experiment, and found that reintroduction of TMEM16K into TMEM16K knockout cells rescues the CTxB trafficking

defect (Fig. 3b), further confirming that TMEM16K is required for proper endosome to trans-Golgi retrograde trafficking.

Lysosomal acidification defect is shown in Fig. 3h. We have done extensive rescue analysis of the lysosomal acidification defect – we tested rescue with wild type TMEM16K, and further tested a mutant TMEM16K with substitutions of calcium-binding acidic residues, as shown in the Fig. 3j. Briefly, TMEM16K mutant cDNA into wild type cells has dominant negative effect. (Please note that we have made a labeling error in our first submission with significance markings, which we have corrected). Furthermore, reintroducing wild type but not mutant TMEM16K rescued the acidification defect of TMEM16K knockout primary cells, demonstrating that TMEM16K is required for normal maturation of the endolysosomal compartments (Fig. 3j).

3. Along the same line, the figure has to be completed with information provided in Suppl. Figure 1 regarding the ER localization. Here as along the entire manuscript, the content of the figures is so condensed that the paper becomes unreadable without having the supplements next to each figure.

Thank you for your helpful comment. We have added TMEM16K localization analysis to Fig 1, as panel Fig. 1e. We have also included quantification of the colocalization of TMEM16K with ER markers Calreticulin, PDI and Sec61 β , as well as colocalization analysis of Sec61 β with Calreticulin and PDI for comparison (Fig. 1f). We wholeheartedly agree that our figures are dense, but with so much data we are not sure how to better represent and include all these experiments.

4. Figure 4a is impossible to read. There is not cell outline and I am just completely puzzled what to look at.

Thank you for the helpful comment. We have repeated the experiment to include the full view of the entire cells, and marked incepts showing the regions presented at higher magnification. We hope that this clarifies what this figure presents (Fig. 4a). Briefly, in this experiment, we have transfected COS7 cells with myc-BioID-TMEM16K, the same construct used for proximity biotinylation proteomics, with the goal to visualize endosomes labeled with biotin in a TMEM16K-dependent manner. We incubated the cells for 6h with biotin, and immunostained for endogenous Rab7 and fluorescently labeled Streptavidin. We show TMEM16K, and overlap of TMEM16K-dependent proximity biotinylation with Rab7. We have also added a line scan for a quantitative analysis.

5. Figure 4g suggests that TMEM16K has a PI-3-P binding domain. This needs to be repeated with recombinant proteins and a flotation assay. I agree that this is indicative, yet it remains incomplete. In addition, the identified PIP-binding deficient mutant needs to be analyzed in their complementation (Figure 3a, b, e) and interaction analyses (Figure 4d).

Thank you for the great suggestion. We have generated N-terminal deletion mutant of TMEM16K that lack amino acids 1-169 (Fig. 5d). This TMEM16K N-terminal deletion mutant properly localized to endoplasmic reticulum and could still reconstitute split-GFP with Rab7, demonstrating that the N-terminal domain is not required for contacts with endosome (Fig. 5e). However, this N-terminal deletion mutant was not able to rescue

endosomal retrograde transport defect when reintroduced in the TMEM16K knockout cells (Fig. 5f,g), showing that the N-terminal domain is required for TMEM16K function.

6. The authors could here also use a complementation assay by mutating the PIP-binding domain and by fusing TMEM16K to a PI-3-P interacting domain. If their interpretation is correct, this should rescue the mutant (if it indeed has a defect in endosomal sorting).

That you for this suggestion. We have generated TMEM16K chimera with P40PX PtdIns3P binding domain instead of TMEM16K N-terminal domain. However, the chimera seemed to form clusters in the ER, indicative of compromised folding of the chimeric protein.

7. Figure 5: The authors suggest that scramblase activity of TMEM16K is important for endosomal transport. However, their mutants in Figure 5 are not analyzed for deficient scramblase activity. This is particularly important for the parts where entire segments are swapped.

We completely agree with the reviewer - we have not directly evaluated scramblase activity in TMEM16K mutants. To do so we would need to purify these mutants as well as wild type TMEM16K as positive control, reconstitute them in liposomes and perform scrambling assays *in vitro* using fluorescent lipids. While these experiments are important for understanding the mechanism of TMEM16K scrambling activity, we feel they are outside of the scope of this study. Based on the crystal structure Bushnell et al. reported in *Nature Communications* (2019), all three single point mutants evaluated in our study are predicted to have impaired scrambling function. The established approach of swapping minimal scrambling domains between family members that are localized in intracellular compartment, like TMEM16K, with TMEM16A or TMEM16F that reside on the cell membrane where scrambling can easily be assayed, have demonstrated the usefulness of domain swapping in assessing scrambling activity (Gyobu et al., 2017; Yu et al., 2015). While TMEM16K scrambling activity is not the main message of the study and we have not directly evaluated scrambling activity of the TMEM16K mutants, considering the aforementioned points we feel our results implicating TMEM16K scrambling activity in endosomal retrograde sorting are of interest to the scientific community. We have adjusted the language to ensure we are precisely describing our findings.

8. I normally would expect a detailed ultrastructural analysis of endosome-ER contact sites and their alterations due to TMEM16K mutants. Has this been done?

We have not done EM analysis. Instead of pursuing EM analysis that entails a significant amount of effort and expertise, we have used a broad range of approaches, including proximity biotinylation proteomics, confocal/live imaging/super-resolution microscopy, split-GFP reconstitution, biochemistry, and a battery of cellular assay and plethora of mutants to substantiate our findings.

References:

- Boncompain, G., Divoux, S., Gareil, N., de Forges, H., Lescure, A., Latreche, L., Mercanti, V., Jollivet, F., Raposo, G., and Perez, F. (2012). Synchronization of secretory protein traffic in populations of cells. *Nat. Methods* 9, 493–498.
- Bonifacino, J.S., and Rojas, R. (2006). Retrograde transport from endosomes to the trans-Golgi network. *Nat. Rev. Mol. Cell Biol.* 7, 568–579.
- Brunner, J.D., Lim, N.K., Schenck, S., Duerst, A., and Dutzler, R. (2014). X-ray structure of a calcium-activated TMEM16 lipid scramblase. *Nature* 516, 207–212.
- Bushell, S.R., Pike, A.C.W., Falzone, M.E., Rorsman, N.J.G., Ta, C.M., Corey, R.A., Newport, T.D., Christianson, J.C., Scofano, L.F., Shintre, C.A., et al. (2019). The structural basis of lipid scrambling and inactivation in the endoplasmic reticulum scramblase TMEM16K. *Nat. Commun.* 10, 1–16.
- Chen, Y., Gershlick, D.C., Park, S.Y., and Bonifacino, J.S. (2017). Segregation in the Golgi complex precedes export of endolysosomal proteins in distinct transport carriers. *J. Cell Biol.* 216, 4141–4151.
- Davis-Kaplan, S.R., Ward, D.M., Shiflett, S.L., and Kaplan, J. (2004). Genome-wide Analysis of Iron-dependent Growth Reveals a Novel Yeast Gene Required for Vacuolar Acidification. *J. Biol. Chem.* 279, 4322–4329.
- Fischer, M.A., Temmerman, K., Ercan, E., Nickel, W., and Seedorf, M. (2009). Binding of plasma membrane lipids recruits the yeast integral membrane protein Ist2 to the cortical ER. *Traffic Cph. Den.* 10, 1084–1097.
- Guyenet, S.J., Furrer, S.A., Damian, V.M., Baughan, T.D., La Spada, A.R., and Garden, G.A. (2010). A Simple Composite Phenotype Scoring System for Evaluating Mouse Models of Cerebellar Ataxia. *J. Vis. Exp. JoVE.*
- Gyobu, S., Ishihara, K., Suzuki, J., Segawa, K., and Nagata, S. (2017). Characterization of the scrambling domain of the TMEM16 family. *Proc. Natl. Acad. Sci. U. S. A.* 114, 6274–6279.
- Hatzipetros, T., Kidd, J.D., Moreno, A.J., Thompson, K., Gill, A., and Vieira, F.G. (2015). A Quick Phenotypic Neurological Scoring System for Evaluating Disease Progression in the SOD1-G93A Mouse Model of ALS. *J. Vis. Exp. JoVE.*
- Jha, A., Chung, W.Y., Vachel, L., Maleth, J., Lake, S., Zhang, G., Ahuja, M., and Muallem, S. (2019). Anoctamin 8 tethers endoplasmic reticulum and plasma membrane for assembly of Ca²⁺ signaling complexes at the ER/PM compartment. *EMBO J.* 38, e101452.
- Kim, A.Y., Tang, Z., Liu, Q., Patel, K.N., Maag, D., Geng, Y., and Dong, X. (2008). Pirt, a phosphoinositide-binding protein, functions as a regulatory subunit of TRPV1. *Cell* 133, 475–485.
- Lalonde, R., and Strazielle, C. (2011). Brain regions and genes affecting limb-clasping responses. *Brain Res. Rev.* 67, 252–259.

Lu, Y., Dong, S., Hao, B., Li, C., Zhu, K., Guo, W., Wang, Q., Cheung, K.-H., Wong, C.W., Wu, W.-T., et al. (2014). Vacuolin-1 potently and reversibly inhibits autophagosome-lysosome fusion by activating RAB5A. *Autophagy* 10, 1895–1905.

Malvezzi, M., Chalal, M., Janjusevic, R., Picollo, A., Terashima, H., Menon, A.K., and Accardi, A. (2013). Ca²⁺-dependent phospholipid scrambling by a reconstituted TMEM16 ion channel. *Nat. Commun.* 4, 2367.

Manford, A.G., Stefan, C.J., Yuan, H.L., Macgurn, J.A., and Emr, S.D. (2012). ER-to-plasma membrane tethering proteins regulate cell signaling and ER morphology. *Dev. Cell* 23, 1129–1140.

McKnight, N.C., Zhong, Y., Wold, M.S., Gong, S., Phillips, G.R., Dou, Z., Zhao, Y., Heintz, N., Zong, W.-X., and Yue, Z. (2014). Beclin 1 Is Required for Neuron Viability and Regulates Endosome Pathways via the UVRAG-VPS34 Complex. *PLOS Genet.* 10, e1004626.

Neefjes, J., and van der Kant, R. (2014). Stuck in traffic: an emerging theme in diseases of the nervous system. *Trends Neurosci.* 37, 66–76.

Pelz, T., Drose, D.R., Fleck, D., Henkel, B., Ackels, T., Spehr, M., and Neuhaus, E.M. (2018). An ancestral TMEM16 homolog from *Dictyostelium discoideum* forms a scramblase. *PLOS ONE* 13, e0191219.

Pi, H., Li, M., Tian, L., Yang, Z., Yu, Z., and Zhou, Z. (2017). Enhancing lysosomal biogenesis and autophagic flux by activating the transcription factor EB protects against cadmium-induced neurotoxicity. *Sci. Rep.* 7, 43466.

Sandvig, K., and van Deurs, B. (2002). Membrane traffic exploited by protein toxins. *Annu. Rev. Cell Dev. Biol.* 18, 1–24.

Scorrano, L., Matteis, M.A.D., Emr, S., Giordano, F., Hajnóczky, G., Kornmann, B., Lackner, L.L., Levine, T.P., Pellegrini, L., Reinisch, K., et al. (2019). Coming together to define membrane contact sites. *Nat. Commun.* 10, 1–11.

Seaman, M.N.J., Michael McCaffery, J., and Emr, S.D. (1998). A Membrane Coat Complex Essential for Endosome-to-Golgi Retrograde Transport in Yeast. *J. Cell Biol.* 142, 665–681.

Sleigh, J.N., Burgess, R.W., Gillingwater, T.H., and Cader, M.Z. (2014). Morphological analysis of neuromuscular junction development and degeneration in rodent lumbrical muscles. *J. Neurosci. Methods* 227, 159–165.

Spinosa, M.R., Progida, C., Luca, A.D., Colucci, A.M.R., Alifano, P., and Bucci, C. (2008). Functional Characterization of Rab7 Mutant Proteins Associated with Charcot-Marie-Tooth Type 2B Disease. *J. Neurosci.* 28, 1640–1648.

Wang, C., Telpoukhovskaia, M.A., Bahr, B.A., Chen, X., and Gan, L. (2018). Endo-lysosomal dysfunction: a converging mechanism in neurodegenerative diseases. *Curr. Opin. Neurobiol.* 48, 52–58.

Yang, H., Kim, A., David, T., Palmer, D., Jin, T., Tien, J., Huang, F., Cheng, T., Coughlin, S.R., Jan, Y.N., et al. (2012). TMEM16F Forms a Ca²⁺-Activated Cation Channel Required for Lipid Scrambling in Platelets during Blood Coagulation. *Cell* 151, 111–122.

Yu, K., Whitlock, J.M., Lee, K., Ortlund, E.A., Cui, Y.Y., and Hartzell, H.C. (2015). Identification of a lipid scrambling domain in ANO6/TMEM16F. *ELife* 4, e06901.

Reviewers' comments:

Reviewer #1 (Remarks to the Author):

The authors have done a great job responding to my previous comments. This paper makes a very significant contribution to the field and opens new avenues for exploration. There remain some hiccups to be addressed.

Overlay in middle panels of Fig. 3c-e are not registered correctly with the images.

Also, the cartoons in d, e are identical despite the reporters being targeted to different compartments.

In Fig. 3g and i, the x-axis is screwy. $100 = \log(\text{sec})$ is a very long time and the text implies that the last data point in Fig. 3g is 60 min (=3600 sec?)

Last line page 4: wild type is repeated twice

Fig. 4 legend. What is an incept?

Fig. 4b. The plot of relative fluorescence intensity seems inverted on the x-axis compared to the adjacent image. Also, the line in the image that indicates the region used to create the plot is offset (does not seem to intersect the EGF signal at all). The x-axis in the plot should be in μm rather than pixels and there should be ticks on the axis. The same applies to other panels in this figure.

Bottom of page 8 refers to the non-scramblase 16K-16A chimera in Fig. 6g ("However the putative scramblase chimera TMEM16K-SCRD16F was able to reconstitute split-GFP with Rab7 (Fig. 6g) and"), but this figure shows point mutants.

Reviewer #2 (Remarks to the Author):

The authors have tried to respond to the reviewer comments. The paper should be published although some of the conclusions are still too strong. This reviewer is not convinced that Golgi structure is normal in the KO cells and the authors should be reminded that MPRs go transiently to the Golgi but at steady state are found in perinuclear late endosomes next to the Golgi. Yes the phenotype is consistent with a block in that overall pathway, and may be due to mis-targeting of enzymes needed for late endosome/lysosome function. Overall, the work is of high quality and interesting.

Reviewer #3 (Remarks to the Author):

The authors provide an almost complete answer to my questions, yet failed to show that the postulated PI-3-P binding resides on very incomplete analyses. I understand that their replacement assays with a PX fusion failed due to misbehavior of the protein, though their claim of PI-3-P specificity then still rests just on the PIP-strip analysis (which is stated in their abstract). I find this too weak to be placed to prominently. Either they tune their statement down or show additional evidence to support their claim.

Likewise, an ultrastructural analysis is normally part of such a study. Finally, I would strongly encourage the authors to include a working model to complete the study.

Response to reviewers' comments

We thank the reviewers for their helpful and constructive comments throughout both rounds of revisions. We have significantly improved our paper by performing several additional experiments and we have modified the figures and the text of the manuscript accordingly. With the new data, we have confirmed as well as extended our conclusions from the original submission. We believe that we have addressed the reviewers' comments and have substantially improved the paper. We hope that the reviewers will now find our revised paper suitable for *Nature Communication*.

The following is a brief summary of the major changes made throughout both rounds of revisions and is followed by our point-by-point response to reviewers' comments.

- We have evaluated 3D morphology of *cis*-Golgi and *trans*-Golgi complex visualized with immunolabeling of endogenous GM130 or TGN38, in acquired 3D reconstructed confocal images, in the wild type and TMEM16K knockout cells (Supplemental Fig. 3b,c)
- We have performed transmission electron microscopy to carry out ultrastructural analysis of ER-endosome membrane contact sites in wild type and TMEM16K knockout cells (Supplemental Fig. 5a,b)
- We have bolstered our ultrastructural analysis with proximity ligation assay with VAPB and Rab7 as markers of ER and endosomes, respectively, in the wild type and TMEM16K knockout cells (Supplemental Fig. 5c,d)
- We have done colocalization analysis of TMEM16K with ER markers Calreticulin, PDI and Sec61 β . We have also included colocalization analysis of Sec61 β with other ER-markers to provide meaningful context for TMEM16K colocalization analysis, as Sec61 β is a pore forming subunit of the translocon complex localized exclusively to ER (Fig. 1e,f)
- We have evaluated whether reintroducing TMEM16K into TMEM16K knockout cells rescues the endosomal retrograde trafficking defect observed with cholera toxin B (CtxB) (Fig. 3b).
- We have evaluated the anterograde biosynthetic pathway using the "Retention Using Selective Hooks" (RUSH) system. RUSH allows synchronization of protein transport through the biosynthetic pathway (Boncompain et al., 2012) and we tracked three transmembrane proteins with different steady state distributions: the glycosylphosphatidylinositol anchor (GPI; transported to plasma membrane), the transferrin receptor (TfR; transported to plasma membrane, early endosomes, and recycling endosomes), and the cation-dependent mannose-6-phosphate receptor (CD-MPR; transported from TGN directly to early/late endosomes). (Fig. 3c-e).
- We have examined EGF colocalization with Rab7 with longer EGF incubation time points (40 min and 60 min). (Fig. 3g)
- We have redone the imaging experiments evaluating TMEM16K proximity to endosomes to include the view of the entire cells, region of interest as well as high magnification insets. We have also included line scan quantifications. (Fig. 4a-e)
- We have included live imaging of TMEM16K with endosomes labeled with Rab7 and EGF. (Fig. 4b, Supplemental Video 3)
- To further evaluate the specificity of TMEM16K interaction with Rab7, we generated Rab7 mutants: constitutively active Q67L, which mimics permanently GTP-bound Rab7, and inactive T22N, which mimics permanently GDP-bound Rab7. We have tested their ability to reconstitute split-GFP with TMEM16K (Fig. 4e,j).
- We have generated N-terminal truncation of TMEM16K, and evaluated its ability to reconstitute split-GFP with Rab7, as well as its ability to rescue endosomal retrograde trafficking defect (Fig. 5d,e,f).

- We have tested effects of pharmacological inhibition of PikFYVE kinase in the wild type and TMEM16K knockout cells. (Fig. 6c,d)
- We have evaluated human disease point mutants and scrambling domains chimeras for their ability to reconstitute split-GFP with Rab7. (Fig. 6g,h)
- We have done RT-PCR analysis evaluating the presence of TMEM16K mRNA in liver and brain tissues obtained from the wild type and TMEM16K ubiquitous knockout littermates. (Supplemental Fig. 1b)

Point by point response to Reviewer Comments, 2nd resubmission:

Reviewer #1 (Remarks to the Author):

The authors have done a great job responding to my previous comments. This paper makes a very significant contribution to the field and opens new avenues for exploration. There remain some hiccups to be addressed. Overlay in middle panels of Fig. 3c-e are not registered correctly with the images.

The Fig.3c-e shows RUSH assay with 3 different RUSH constructs. Columns are different time points after addition of biotin, showing distribution of the RUSH construct at that time point. First row of images shows scheme of the distribution of the RUSH construct at that corresponding time point, second row shows representative image of the wild type cells, and third row shows representative images of the TMEM16K knockout cells. There are no overlays shown in these panels. Thank you for pointing out how this layout is confusing; we have clarified the figure panels and text.

Also, the cartoons in d, e are identical despite the reporters being targeted to different compartments.

The cartoons are different, but as you kindly pointed out, with far too subtle colors and line thickness to be helpful. We have improved the contrast and clarity of the schemes.

In Fig. 3g and i, the x-axis is screwy. $100 = \log(\text{sec})$ is a very long time and the text implies that the last data point in Fig. 3g is 60 min (=3600 sec?)

The time points for the EGF-Alexa647 pulse-chase experiment are 10 min, 15 min, 40 min and 60 min. We have corrected the scale label to minutes, thank you for noticing this error.

Last line page 4: wild type is repeated twice

Thank you, we have corrected it.

Fig. 4 legend. What is an incept?

It should have been written as inset. We have corrected the misspelling.

Fig. 4b. The plot of relative fluorescence intensity seems inverted on the x-axis compared to the adjacent image. Also, the line in the image that indicates the region used to create the plot is offset (does not seem to intersect the EGF signal at all). The x-axis in the plot should be in μm rather than pixels and there should be ticks on the axis. The same applies to other panels in this figure.

Thank you for the helpful observation, we have improved the figures.

Bottom of page 8 refers to the non-scramblase 16K-16A chimera in Fig. 6g ("However the putative scramblase chimera TMEM16K-SCRD16F was able to reconstitute split-GFP with Rab7 (Fig. 6g) and"), but this figure shows point mutants.

Thank you, we have corrected the figure assignment.

Reviewer #2 (Remarks to the Author):

The authors have tried to respond to the reviewer comments. The paper should be published although some of the conclusions are still too strong. This reviewer is not convinced that Golgi structure is normal in the KO cells and the authors should be reminded that MPRs go transiently to the Golgi but at steady state are found in perinuclear late endosomes next to the Golgi. Yes the phenotype is consistent with a block in that overall pathway, and may be due to mis-targeting of enzymes needed for late endosome/lysosome function. Overall, the work is of high quality and interesting.

Thank you for this valid point. Experiments using cholera toxin corroborated our findings of impaired endosomal retrograde transport obtained with pulse chase experiment with CI-MPR. We have looked at the distribution of multiple Golgi resident proteins in wild type and TMEM16K knockout cells, and found no obvious difference. We have evaluated the trafficking pathway of three transmembrane proteins with different steady state distributions, which all pass through Golgi, and found no difference between wild type and TMEM16K knockout cells. We have now included a more detailed evaluation of 3D Golgi morphology. We have performed immunocytochemistry to label *cis* or *trans*-Golgi in wild type and TMEM16K knockout cells, acquired dense z-stacks on the confocal microscope and reconstructed the corresponding Golgi structures in 3D using Imaris Software. We have found no difference in volume, area nor index of fragmentation (defined as volume/area) between the wild type and TMEM16K knockout cells. However, we agree with the reviewer that we cannot exclude the possibility of a more subtle perturbances to the Golgi complex, and indeed, we would whole-heartedly agree that in the absence of TMEM16K some proteins needed for the late endosome/lysosome function could be mistargeted due to the observed defect in endosome sorting. We have included this sentence in the discussion to provide further clarification.

Reviewer #3 (Remarks to the Author):

The authors provide an almost complete answer to my questions, yet failed to show that the postulated PI-3-P binding resides on very incomplete analyses. I understand that their replacement assays with a PX fusion failed due to misbehavior of the protein, though their claim of PI-3-P specificity then still rests just on the PIP-strip analysis (which is stated in their abstract). I find this too weak to be placed to prominently. Either their tune their statement down or show addition evidence to support their claim.

We have adjusted the text to more precisely communicate our results.

Likewise, an ultrastructural analysis is normally part of such a study.

We have combined in this study almost all experimental approaches so far employed to study membrane contacts (Scorrano et al., 2019). We have now included the ultrastructural analysis to evaluate the membrane contact sites between ER and endosomes, and their possible alterations in the absence of TMEM16K. Using electron microscopy as previously reported (Kilpatrick et al., 2017), we found no difference in the percentage of endosomes in close proximity (>30 nm) of ER, between wild type and

TMEM16K knockout cells. We have further bolstered these observations with proximity ligation assay (PLA) *in situ*, a powerful novel approach to study contact sites alterations in a highly quantitative manner (Lim et al., 2019). Using PLA *in situ* with VAPB and Rab7 as markers of ER and endosomes, respectively, we found no difference in the extent of ER-endosome MCS, as measured by the number of PLA puncta, between the wild type and TMEM16K knockout cells. These are not unexpected results, as most contacts have multiple tethering molecules so deleting any singular tether is unlikely to eliminate a contact. One of the classical examples is deletion of the TMEM16K yeast homolog, Ist2, which does not significantly affect the extent of ER-PM contact (Manford et al., 2012; Toulmay and Prinz, 2011). Additional concomitant deletion of the other 5 ER-PM MCS proteins, namely three tricalbins (homologs to mammalian Extended Synaptotagmins), Scs2 (homolog to mammalian VAPA) and Scs22 (homolog to mammalian VAPB), was required for strong reduction of ER-PM contact. Likewise, multiple distinct contact sites between the ER and endosomes have been identified to date, with diverse molecular compositions (Raiborg et al., 2015a). Further supporting the possibility of redundancy, our TMEM16K proteomics dataset includes multiple proteins known to function at ER-endosomal contact sites including VAPA and VAPB (Alpy et al., 2013; Dong et al., 2016; Rocha et al., 2009), SNX1 and SNX2 (Dong et al., 2016), Rab7A (Friedman et al., 2013; Raiborg et al., 2015b; Rocha et al., 2009), and PTP1B (Eden et al., 2010) (Fig. 2b; Supplemental Table 1).

Besides proximity between two membranous organelles, a *bona fide* membrane contact site (MCS) should fulfill specific functions. Most MCS proteins have not only structural role, but also non-mutually exclusive functional and/or regulatory roles (Eisenberg-Bord et al., 2016; Scorrano et al., 2019). We found TMEM16K mutants which can reconstitute split-GFP with Rab7, but are unable to rescue the CI-MPR retrograde trafficking defect in TMEM16K knockout cells (Fig. 5d,e, Fig. 6g,i,j), so these mutants can still form contacts with endosomes, but cannot fulfill the TMEM16K cellular function. That would suggest that TMEM16K has functional and/or regulatory role at contact sites with endosomes enabling endosomal sorting, and our results further raise the possibility that the functional or regulatory role of TMEM16K could involve phospholipid scrambling.

Finally, I would strongly encourage the authors to include a working model to complete the study.

Thank you for this kind encouragement. We have expanded on our hypotheses to spell out how our data and the literature could fit in all together, hopefully providing a testable framework to be evaluated in the future.

Point by point response to Reviewer Comments, 1st resubmission:

Reviewer #1 (Remarks to the Author):

This paper from the Jan lab provides evidence that TMEM16K plays a role in endosomal transport and suggests that mutations in TMEM16K may produce SCAR10 as a result of defects in endosomal sorting. This is an extremely interesting, but sprawling, investigation that opens new territory in both the TMEM16 and membrane trafficking fields. While I am highly intrigued by its novelty, overall I think that the conclusions are drawn in a somewhat cavalier manner from data that is preliminary or incomplete.

Major issues:

There is no characterization of the knockout mouse to show that it does not express TMEM16K.

We have added RT-PCR results showing the absence of TMEM16K mRNA in TMEM16K knockout mice (Supplemental Fig. 1b). Information about all the primers for RT-PCR as well as genotyping are in the Material and Methods, along with the identification information from the International Knockout Mouse Consortium that generated the TMEM16K conditional knockout mice used in our study.

The authors begin by showing that the TMEM16K knockout mice exhibit hindlimb clasping and smaller neuromuscular junctions, but the paper does not provide an explanation. While the implication that this phenotype can be explained by defective endosomal trafficking is appealing, there are no data to allow such a conclusion.

While we would very much like to provide direct demonstration that the observed defects in endosomal transport are the cause of the pathology, achieving this goal would require demonstration that the observed cellular defects can be rescued in a way different from reintroducing TMEM16K to mutant cells (for example with small molecule) and the same treatment can also relieve the pathology. While no such treatment has been identified, we have tested the converse scenario, and shown that mutations of three different single amino acids in TMEM16K that cause the pathology in humans also severely impact the ability of these TMEM16K mutants to rescue the observed cellular defects (Fig. 6g,i,j).

The proteomic analysis in Fig. 2 and S-Table 1 is not well presented, confusing, and appears misleading. S-Table 1 shows <50 proteins, but Fig. 2 shows 365 proteins. The correspondence between the table and figure is inscrutable and it is not clear how much of Fig. 2 is actual data and how much is bioinformatics. For example, of the 16 proteins in S-Table 1 listed as Endosomal transport, only 5 appear in the pink panel labelled Endosomal Transport in Fig. 2c. The majority of the proteins in this panel do not appear in S-Table 1.

Thank you for this helpful comment. We have clarified the text.

Briefly, Fig. 2b shows actual raw data – the obtained candidates from the TMEM16K proteomics. To identify in an unbiased manner the most biologically relevant functional clusters which could infer TMEM16K function rather than just hand-picking one or two highly ranked candidates for our study, we have visualized the list of these candidates as protein-protein interaction network using String app in the Cytoscape.

From here, we have done 2 bioinformatics analysis:

(1) We identified functional enrichment categories based on GO terms, which we overlaid with color-code on our list of candidates. Indeed, visualization of this overlay suggests functional clusters of proteins identified in the TMEM16K proteomics.

(2) We performed a mathematical clustering analysis using MCODE cluster app in Cytoscape to identify clusters from our raw data, generating the simplified network shown in Fig. 2c. Clusters are defined as highly interconnected nodes in the network. We then overlaid the color-coding from functional enrichment analysis on this simplified network generated via clustering analysis.

We have originally listed only a subset of the proteins listed in the Supplemental Table 1 to indicate their cellular functions. Thank you for your helpful comment to point out that this is confusing - we have added the full list of candidates to Supplemental Table 1. In addition, all the raw proteomics data will be made available to other researchers.

Fig. 3a,b shows nicely that the trafficking of CI-MPR and CtxB are altered in the knockout, but does this mean that endosome-to-Golgi trafficking is reduced, or that retention in the Golgi is reduced possibly by increased anterograde traffic? Experiments to sort this out are necessary to justify the conclusion that “TMEM16K is required for proper endosome to trans-Golgi retrograde trafficking”.

CI-MPR and CTxB are classical markers used to evaluate endosome to TGN retrograde trafficking by performing pulse chase experiments, so in these experiments we are following only the retrogradely trafficked molecules from the plasma membrane through endosomes to TGN. Their mis-localization in pulse chase experiments shows that the endosome to trans-Golgi retrograde trafficking is perturbed as previously reported in literature (Bonifacino and Rojas, 2006; Sandvig and van Deurs, 2002; Seaman et al., 1998). However, as this reviewer pointed out, one can imagine a scenario in which anterograde secretory pathway is perturbed upstream of retrograde trafficking from endosomes to TGN. To address this question, we took advantage of recent methodological developments that allow synchronization of protein transport through the biosynthetic pathway (Chen et al., 2017). Using the “Retention Using Selective Hooks” (RUSH) system we tracked the biosynthetic transport of three transmembrane proteins with different steady state distributions: the glycosylphosphatidylinositol anchor (GPI; transported to plasma membrane), the transferrin receptor (TfR; transported to plasma membrane, early endosomes, and recycling endosomes), and the cation-dependent mannose-6-phosphate receptor (CD-MPR; transported from TGN directly to early/late endosomes). We found no difference in the transport through the biosynthetic pathway between the MEF cells from wild type mice and TMEM16K knockout mice, showing that the anterograde secretory pathway is unaffected (Fig. 3c,d,e).

The localization of TMEM16K in ER is not quantitatively established. Although the data show that the bulk of it is in ER, expression in the endolysosomal system at a lower level is not investigated rigorously.

We have shown ER localization in both live imaging and immunofluorescence (Fig. 1e, Supplemental Video 2). We have quantified the colocalization of TMEM16K with Calreticulin, PDI and Sec61 β (Fig. 1f). We have also done immunohistochemistry and quantified in the same manner the colocalization of Sec61 β with Calreticulin and PDI to provide a meaningful context for the colocalization analysis obtained with TMEM16K (Fig. 1f), given that Sec61 β is a pore forming component of the translocon complex localized exclusively to the ER.

Additionally, Bushnell et al. reported in *Nature Communication* (2019) TMEM16K crystal structure as well as an analysis of the TMEM16K localization to the ER, further corroborating our findings.

The authors’ interpretation of the split GFP experiments assumes that 16K is exclusively localized in ER and Rab7 exclusively in endosomes. If even a small fraction of 16K and Rab7 were localized in the same compartment, one might expect to see reconstitution of fluorescence.

Thank you for raising this point. We have tested several other endosomal proteins found in our proteomics (OSBPL11, SNX1, SNX2, VPS26) with the split-GFP assay (Supplemental Fig. 4) and did not observe positive GFP signal with any of the other tested proteins besides Rab7 (Fig. 4h), so it seems unlikely that a small fraction of the TMEM16K is in the endosomal compartment. We have also found that TMEM16K mutants (F337V, D615N, SCRDA), which are all localized in the ER, are not able to

reconstitute the split-GFP with Rab7 (Fig. 6g,h), further supporting the specificity of this interaction. Likewise, TMEM16K was able to reconstitute split-GFP only with the constitutively active Rab7 mutant Q67L (Fig. 4i), but not with the inactive T22N Rab7 mutant (Fig. 4j), again corroborating the specificity of the interaction.

The authors do not show why lysosomal pH is altered. Does it alter the trafficking of lysosomal channels that regulate pH (like the v-H-ATPase)? If so, this should be tested. If 16K is present in lysosomes and, like other TMEM16s, conducts ions, it might alter lysosomal pH directly. More information is required to allow the conclusion that (Page 4) “these results show that loss of TMEM16K causes a defect in endosomal retrograde sorting, which can cause deficiencies within the later stages of the endolysosomal system”.

As pointed out by the reviewer, V-ATPase and other lysosomal channels indeed could have perturbed trafficking, and this will be the focus of follow-up studies. However, it does not seem likely TMEM16K is directly contributing to altering lysosomal pH given that TMEM16K was shown to be a scramblase and we and others have localized it to the endoplasmic reticulum (Bushell et al., 2019). We have revised the statement.

I do not know how to interpret Fig. 4a and b. The bright SA labeling in Fig. 4a seems to enclose a large cluster of Rab7 positive structures (~2 um in diameter) and colocalization of 16K and Rab7 seems minimal. But in Fig. 4b, 16K and Rab7 co-localize perfectly, which is not what one would expect if ER and endosomes are interacting. A more quantitative analysis is required.

Thank you for these helpful comments. To address these comments, we have included live imaging showing fluorescently tagged TMEM16K, with fluorescently tagged Rab7 and EGF-Alexa647 (Fig. 4b, Supplemental Video 3). These experiments reveal highly dynamic movements of TMEM16K and Rab7, as well as their interactions. We have also added line scans for a quantitative analysis.

The conclusion that “ER-localized TMEM16K forms contact sites with endosomes, interacting with Rab7 and PtdIns(3)P” is not justified. The authors only show that PI3P binds to 16K N-terminus in an *in vitro* system and that knockout of TMEM16K alters the size of PI3P-positive vesicles.

We have used a broad range of approaches to support our conclusion that TMEM16K forms contacts with Rab7-endosomes, including proximity biotinylation proteomics (Fig. 2), confocal/live imaging/super-resolution microscopy (Fig. 4), and split-GFP reconstitution (Fig. 4, 5, 6), by following precedence in the literature (Scorrano et al., 2019). We have validated the specificity of this interaction using mutants of both Rab7 and TMEM16K (Fig. 4i,j, 6g,h). We have further shown *in vitro* binding of a subdomain of TMEM16K to a subset of phosphatidylinositols, with two other mammalian members used as control (Fig. 5a-c), as done for various phosphatidylinositol binding proteins (Fischer et al., 2009; Kim et al., 2008). We have also found that the absence of TMEM16K leads to enlarged PI3P-vesicles, and showed that reintroducing TMEM16K can rescue the defect (Fig. 6b,d), as it was similarly done for Ist2 and PtdIns4P at the plasma membrane (Fischer et al., 2009; Manford et al., 2012). We have modified the wording to more precisely represent our findings: “TMEM16K forms contact sites with endosomes, reconstituting split-GFP with small GTPase Rab7 and binding phosphatidylinositol 3-phosphate (PtdIns(3)P).”

Specific comments.

Because there is some variability in the literature, the authors should state exactly what the box plots mean.

We have added the following clarification to the Statistics section in Materials and Methods: “We used box plot to graphically visualize data where the box includes the first quartile and the third quartile, with the central line representing the median. Whiskers represent the minimum and maximum values of data. X inside the box represents the mean of data.”

Page 2. “While the yeast homolog Ist2 forms MCS that play a vital role in lipid homeostasis at MCS between the ER and plasma membrane. “ IST2 has not been shown to participate in lipid homeostasis, to my knowledge.

Manford et al. *Developmental Cell* (2012) reported the accumulation of plasma membrane PtdIns4P in the absence of Ist2. Such accumulation leads to perturbed exchange of lipids at the ER-plasma membrane MCS. We have included this citation in the revised manuscript.

Page 2. “mammalian family member most closely related to Ist2.” I think that this is an exaggeration. The mammalian homologs have only 5 – 9% identity to Ist2. Although TMEM16K may have the highest percent identity (9%), any implication that TMEM16K has any functional similarity to Ist2 is unwarranted. Fig. 1a should have a scale bar and bootstrap values.

While 9% is low for amino acid identity, this was what sparked our interest to pursue the question whether TMEM16K might be involved in membrane contact sites. With a similarly low percentage of amino acid identity between TMEM16 family members, there is functional similarity among lipid scramblases such as mammalian TMEM16K (Bushell et al., 2019), mammalian TMEM16F (Yang et al., 2012), fungal afTMEM16 (Malvezzi et al., 2013) and nhTMEM16 (Brunner et al., 2014), as well as a distant ameboid DfTMEM16 homolog (Pelz et al., 2018). Considering that the mammalian TMEM16 family members are modulators of diverse cellular functions, it will be of significant interest and physiological relevance to determine whether some members of this family have a role in interorganelle communication. As we now state in our revised manuscript, since submission of this manuscript, another study was recently published showing that mammalian TMEM16H has a role at membrane contact sites between ER and plasma membrane (Jha et al., 2019).

S-Fig.1c and S-Fig. 2a,b are interpreted too casually. While the colocalization of 16K with ER-tdTomato is convincing, the co-localization with calreticulin and PDI is less obvious. It seems that PDI and 16K are in different sub-compartments even though there is general overlap. A quantitative treatment of these data is necessary. Also, because overexpressed proteins often accumulate in the ER, is there any data that endogenous TMEM16K is concentrated in ER?

Localization of the endogenous proteins compared to tagged version can be an issue. We have tested commercially available antibodies and generated 2 different TMEM16K antibodies to visualize endogenous protein, but we found the antibodies were not specific. Hence, we have used a multitude of different tags and have found that all the constructs yielded only ER localization as assessed by both live imaging and immunofluorescence (Fig. 1e,f, Supplemental Video 2). We have performed quantification of colocalization

between TMEM16K and ER markers Calreticulin, PDI and Sec61 β . We further cite in our revised manuscript the recent publication of Bushell et al. in *Nature Communications* (2019), which reports TMEM16K crystal structure as well as TMEM16K localization to endoplasmic reticulum.

Page 2. “knockout mice displayed increasing hindlimb clasping, a behavioral phenotype characteristic of impaired neuromuscular function.” Please provide a reference. It seems to me that hindlimb clasping is more likely explained by CNS defects rather than defective neuromuscular junctions.

Hindlimb clasping is a behavioral marker of disease progression in a number of mouse models of neurodegeneration, including ataxias, and is characteristic of impaired neuromuscular function (Guyenet et al., 2010; Hatzipetros et al., 2015). As the reviewer pointed out, this impaired neuromuscular function indeed can be due to defects in the CNS including the spinal cord (Lalonde and Strazielle, 2011). We also need to bear in mind that neuromuscular junctions correspond to the synaptic contacts between muscle and axon terminals of motor neurons in the CNS. Reduction of neuromuscular junctions is an early pathological target of impaired neuromuscular function that is easily accessible for histological visualization and quantification (Sleigh et al., 2014). We have included citations in the revised manuscript.

Page 3. “Because dysfunctions of endosomal transport are tightly associated with neurodegenerative diseases.” What does “tightly associated with” mean?

With an increasing number of genes involved in the endolysosomal pathway implicated in neurodegenerative diseases, endolysosomal dysfunction represents a pathophysiological mechanism shared across these diseases (Neefjes and van der Kant, 2014; Wang et al., 2018). We have provided additional references in the revised manuscript.

Fig. 3f legend: (WT slope= -0.633, KO slope= -0.0438). These numbers are in error. There is not a >10-fold difference in -slopes. Also, please provide units. There are no error bars.

Thank you, indeed this is an error. We have corrected the mistake and provided the equations and R^2 .

Fig. 3g. It is not clear how these experiments were performed: is this a steady-state measurement in the absence of FCCP? Because Lysosensor is not ratiometric is it ratioed vs. Cherry-CAAX fluorescence? Are these measurements reliable without a ratiometric indicator?

Thank you for a well raised point. Lysosensor green DNP-189 is extensively used in literature as a reliable way of qualitatively evaluating pH of the acidic compartments (Davis-Kaplan et al., 2004; Lu et al., 2014; McKnight et al., 2014; Pi et al., 2017). We have performed the experiments in the following manner: we loaded Lysosensor Green DNP-189 over 30 min incubation, washed twice with imaging media and all acquisition was done over the next 3 min, where we measured the fluorescence intensity per cell.

We have additionally tested the specificity of the measured signal with Lysosensor Green DNP-189 using proton ionophore FCCP (Fig. 3i) Furthermore, we have performed extensive rescue experiment with both wild type and mutant TMEM16K cDNA (Fig. 3j).

Fig. 4d – Is the box in 16K-V5-GFP11 drawn correctly to show enlarged area?

Thank you, indeed the box marking the inset on the 16K-V5-GFP11 image was shifted. We have corrected it.

Images showing overlay of 3 colors are not useful because there is too much information. I would prefer to see overlays of 2 key channels at a time so that I can clearly see colocalization

We were unsure which two channels would constitute key channels, so we have added the line scan quantification to simplify interpretation of the colocalization (Fig 4a,b,e). We hope this helps in evaluating the represented data.

Reviewer #2 (Remarks to the Author):

There is great interest in the composition and function of membrane contact sites, and this paper has used BioID to identify proteins in the vicinity of TMEM16K, an ER protein that is a lipid scramblase linked to spinocerebellar ataxia. In mouse ubiquitous and neuron specific knock outs, the authors observed reductions in the sizes of neuromuscular junctions and overall impaired neuromuscular function. This was not well connected to the rest of the story.

As mentioned previously, while we would very much like to provide direct demonstration that the observed defects in endosomal transport are the cause of the pathology, this would require that we can rescue the observed cellular defects in a different way (for example with small molecule) so that we can test whether this treatment that rescues the cellular defects also relieve the pathology. Before such a treatment can be developed, we have tested the converse scenario, and shown that three different TMEM16K mutations with single amino acid substitution that cause the pathology in humans also compromise the ability of TMEM16K to rescue the observed cellular defects.

Upon BioID, the authors detected ER and endosomal protein interactions. The authors noted dispersal of mannose 6-phosphate receptors from a perinuclear compartment and decreased transport of cholera toxin to the Golgi as well as weaker endo-lysosomal acidification which could have been due to defects in trafficking of newly synthesized lysosomal enzymes and the V-ATPase. (Note that MPRs usually are seen in perinuclear late endosomes, not the TGN at steady state).

To avoid the potential caveats mentioned by the reviewer, we pursued pulse chase experiments looking at trafficking of internalized CI-MPR from the plasma membrane through endosome to TGN. We have clarified the text that we are looking at the pulse chase experiment at the t = 60 min time point, and not a steady state localization.

The authors note an interaction with Rab7 by split GFP and STED reveals compartment interaction. (A nice control would have been a GDP-locked Rab7).

Thank you for the great suggestion. We have generated inactive GFP(1-10)-HA-T22N Rab7 and constitutively active GFP(1-10)-HA-Q67L Rab7 mutants (Spinosa et al., 2008), and evaluated their ability to reconstitute GFP with TMEM16K-V5-GFP11. We found that TMEM16K was able to reconstitute split-GFP only with the constitutively active Rab7 Q67L

mutant (Fig. 4i), but not with the inactive Rab7 T22N mutant (Fig. 4j), further validating the specificity of the observed contact between TMEM16K and Rab7 endosomes.

Dot blots suggest that the protein binds PI3P but Rab7 is late endosomal where there would be more PI3,5P. Because dot blots aren't perfect, it might be worth trying to recapitulate this interaction in a liposome flotation scenario, although the accumulation of PI3P structures is consistent with what was shown. Interestingly, TMEM16K seems to influence PI3P levels independent of VAPA/B, enlarging these structures. Is it blocking early to late endosome Rab conversion? Transferrin recycling was not altered, nor was EGF co-localization with Rab7. Longer half time of EGF degradation would provide indication of whether conversion was altered and this is easy to add and should be.

Thank you for this great suggestion. We have performed this experiment looking at the 40 min and 60 min time point. We found no difference between the wild type and knockout cells in the colocalization of the EGF with the late endosomal markers Rab7 at 10 min, 15 min and 40 min time points (Fig. 3g), indicating that the mutant phenotype arose from a defect at or after the Rab7 stage of endolysosomal maturation. However, we found greater retention of EGF in Rab7 endosomes at 60 min in TMEM16K knockout cells compared to wild type cells (Fig. 3g), suggesting defect in endosomal sorting.

In summary, there is a lot of nice data here implicating a lipid scramblase in a new class of junction. The precise mechanism is not yet shown but the data will be of interest to workers in this area. Perhaps the authors could look at the consequences of KNOCKOUT on PikFYVE kinase that generates PI3,5P as its activity seems to be lacking

Thank you for another great suggestion. We have tried evaluating PtdIns(3,5)P₂ with the available lipid sensor ML1Nx2, but we were not convinced it could be reliably used for studies of primary mouse fibroblasts. Hence we used pharmacological manipulation with YM201636 to inhibit PikFYVE kinase to evaluate the observed phenotypes in the wild type and TMEM16K knockout cells using the PtdIns3P lipid sensor P40XP-gfp. Inhibiting PIKfyve in TMEM16K wild type cells recapitulated the TMEM16K knockout phenotype. However, in the TMEM16K knockout cells we did not observe additional cumulative effect, suggesting that conversion between PtdIns3P to PtdIns(3,5)P₂ is impaired in the absence of TMEM16K (Fig. 6c,d).

Also, the authors should try to link the mouse phenotypes with the cell biology.

We wholeheartedly agree with the reviewer that it would be great to further investigate how the observed cellular defects lead to the mouse phenotypes. Such an endeavor will require developing new ways to rescue the cellular defects so as to test whether rescue approaches other than reintroducing wildtype TMEM16K, but not TMEM16K bearing disease-causing mutations, can also rescue the mouse phenotypes – a fairly substantial new project. We feel that sharing our findings with the community at this point before embarking another demanding project will be of interest; our revised manuscript will contribute to the better understanding of this link by facilitating the development of follow up studies by our lab as well as our colleagues.

Reviewer #3 (Remarks to the Author):

The manuscript of Petkovic et al. focuses on the role of the lipid scramblase TMEM16K as a potential regulator of endosomal sorting. The authors demonstrate that TMEM16K contact proteins of the endosomal machinery, likely via contact sites, and interacts with Rab7 and PI-3-P. Loss of TMEM16K impairs endosomal transport, yet not other trafficking pathways. The authors suggest that TMEM16K is a critical factor in ER-endosome contact site formation and endosomal sorting.

The authors provide a full spectrum of assays to address the critical role of TMEM16K in endosomal biogenesis. Their analysis of the knock-out mutant is overall extensive, and complementation analyses suggest that the scramblase activity is required for TMEM16K function. While I consider the extent of their analysis impressive, I miss several controls to be able to judge the relevance of their observations relative to the loss of known endosomal proteins. Also lipid scramblase activity needs are not really addressed. My detailed points are listed below:

1. Figure 3 is the first figure where defects in TMEM16K are addressed. The authors need to show a positive control for endosomal transport defects (a,b). Otherwise it is not possible to judge the relevance. Is this a strong defect or a mild one? The same applies for Figure 3e-g. There is no knock out analysis shown without complementation with wild-type or mutant 16K, or any other known mutant/siRNA knock out affected in lysosomal biogenesis. This is again critical to judge if TMEM16K is just needed as a supporting factor or has an overall critical role.

We are grateful for your correction. Indeed, the point of our experiments was not to claim that TMEM16K is more or less important than other proteins shown to regulate endosomal retrograde trafficking, but to show that it is involved in this process, and that in the absence of this ER protein endosomal retrograde trafficking is perturbed. We have corrected the text appropriately to avoid misconception. Many labs have done important work studying this pathway, and we feel it would be beyond the scope of this paper to expand it to include hierarchical analysis with other known regulators.

2. Figure 3b does not show complementation with the wild-type protein. This needs to be added. Also Figure 3e is incomplete here.

Thank you for pointing it out. We have added the rescue experiment. We found that reintroduction of TMEM16K into TMEM16K knockout cells rescues the CTxB trafficking defect (Fig. 3b), further confirming that TMEM16K is required for proper endosome to trans-Golgi retrograde trafficking.

Lysosomal acidification defect is shown in Fig. 3h. We have done extensive rescue analysis of the lysosomal acidification defect – we tested rescue with wild type TMEM16K, and further tested a mutant TMEM16K with substitutions of calcium-binding acidic residues, as shown in the Fig. 3j. Briefly, TMEM16K mutant cDNA introduced into wild type cells has dominant negative effect. (Please note that we have made a labeling error in our first submission with significance markings, which we have corrected). Furthermore, reintroducing wild type but not mutant TMEM16K rescued the acidification defect of TMEM16K knockout primary cells, demonstrating that TMEM16K is required for normal maturation of the endolysosomal compartments (Fig. 3j).

3. Along the same line, the figure has to be completed with information provided in Suppl. Figure 1 regarding the ER localization. Here as along the entire manuscript, the content of the figures is

so condensed that the paper becomes unreadable without having the supplements next to each figure.

Thank you for your helpful comment. We have added TMEM16K localization analysis to Fig 1, as panel Fig. 1e. We have also included quantification of the colocalization of TMEM16K with ER markers Calreticulin, PDI and Sec61 β , as well as colocalization analysis of Sec61 β with Calreticulin and PDI for comparison (Fig. 1f). We wholeheartedly agree that our figures are dense, but with so much data we are not sure how to better represent and include all these experiments.

4. Figure 4a is impossible to read. There is not cell outline and I am just completely puzzled what to look at.

Thank you for the helpful comment. We have repeated the experiment to include the full view of the entire cells, and marked insets showing the regions presented at higher magnification. We hope that this clarifies what this figure presents (Fig. 4a). Briefly, in this experiment, we have transfected COS7 cells with myc-BioID-TMEM16K, the same construct used for proximity biotinylation proteomics, with the goal to visualize endosomes labeled with biotin in a TMEM16K-dependent manner. We incubated the cells for 6h with biotin, and immunostained for endogenous Rab7 and fluorescently labeled Streptavidin. We show TMEM16K, and overlap of TMEM16K-dependent proximity biotinylation with Rab7. We have also added a line scan for a quantitative analysis.

5. Figure 4g suggests that TMEM16K has a PI-3-P binding domain. This needs to be repeated with recombinant proteins and a flotation assay. I agree that this is indicative, yet it remains incomplete. In addition, the identified PIP-binding deficient mutant needs to be analyzed in their complementation (Figure 3a, b, e) and interaction analyses (Figure 4d).

Thank you for the great suggestion. We have generated N-terminal deletion mutant of TMEM16K that lack amino acids 1-169 (Fig. 5d). This TMEM16K N-terminal deletion mutant properly localized to endoplasmic reticulum and could still reconstitute split-GFP with Rab7, demonstrating that the N-terminal domain is not required for contacts with endosome (Fig. 5e). However, this N-terminal deletion mutant was not able to rescue endosomal retrograde transport defect when reintroduced in the TMEM16K knockout cells (Fig. 5f,g), showing that the N-terminal domain is required for TMEM16K function.

6. The authors could here also use a complementation assay by mutating the PIP-binding domain and by fusing TMEM16K to a PI-3-P interacting domain. If their interpretation is correct, this should rescue the mutant (if it indeed has a defect in endosomal sorting).

Thank you for this suggestion. We have generated TMEM16K chimera with P40PX PtdIns3P binding domain instead of TMEM16K N-terminal domain. However, the chimera seemed to form clusters in the ER, indicative of compromised folding of the chimeric protein.

7. Figure 5: The authors suggest that scramblase activity of TMEM16K is important for endosomal

transport. However, their mutants in Figure 5 are not analyzed for deficient scramblase activity. This is particularly important for the parts where entire segments are swapped.

We completely agree with the reviewer - we have not directly evaluated scramblase activity in TMEM16K mutants. To do so we would need to purify these mutants as well as wild type TMEM16K as positive control, reconstitute them in liposomes and perform scrambling assays *in vitro* using fluorescent lipids. While these experiments are important for understanding the mechanism of TMEM16K scrambling activity, we feel they are beyond the scope of this study. Based on the crystal structure that Bushnell et al. reported in *Nature Communications* (2019), all three single point mutants evaluated in our study are predicted to have impaired scrambling function. The established approach of swapping minimal scrambling domains between family members that are localized in intracellular compartment, like TMEM16K, with TMEM16A or TMEM16F that reside on the cell membrane where scrambling can easily be assayed, have demonstrated the usefulness of domain swapping in assessing scrambling activity (Gyobu et al., 2017; Yu et al., 2015). While TMEM16K scrambling activity is not the main message of the study and we have not directly evaluated scrambling activity of the TMEM16K mutants, considering the aforementioned points we feel our results implicating TMEM16K scrambling activity in endosomal retrograde sorting are of interest to the scientific community. We have adjusted the language to ensure we are precisely describing our findings.

8. I normally would expect a detailed ultrastructural analysis of endosome-ER contact sites and their alterations due to TMEM16K mutants. Has this been done?

We have not done EM analysis. Instead of pursuing EM analysis that entails a significant amount of effort and expertise, we have used a broad range of approaches, including proximity biotinylation proteomics, confocal/live imaging/super-resolution microscopy, split-GFP reconstitution, biochemistry, and a battery of cellular assay and plethora of mutants to substantiate our findings.

References:

Alpy, F., Rousseau, A., Schwab, Y., Legueux, F., Stoll, I., Wendling, C., Spiegelhalter, C., Kessler, P., Mathelin, C., Rio, M.-C., et al. (2013). STARD3 or STARD3NL and VAP form a novel molecular tether between late endosomes and the ER. *J. Cell Sci.* 126, 5500–5512.

- Boncompain, G., Divoux, S., Gareil, N., de Forges, H., Lescure, A., Latreche, L., Mercanti, V., Jollivet, F., Raposo, G., and Perez, F. (2012). Synchronization of secretory protein traffic in populations of cells. *Nat. Methods* 9, 493–498.
- Bonifacino, J.S., and Rojas, R. (2006). Retrograde transport from endosomes to the trans-Golgi network. *Nat. Rev. Mol. Cell Biol.* 7, 568–579.
- Brunner, J.D., Lim, N.K., Schenck, S., Duerst, A., and Dutzler, R. (2014). X-ray structure of a calcium-activated TMEM16 lipid scramblase. *Nature* 516, 207–212.
- Bushell, S.R., Pike, A.C.W., Falzone, M.E., Rorsman, N.J.G., Ta, C.M., Corey, R.A., Newport, T.D., Christianson, J.C., Scofano, L.F., Shintre, C.A., et al. (2019). The structural basis of lipid scrambling and inactivation in the endoplasmic reticulum scramblase TMEM16K. *Nat. Commun.* 10, 1–16.
- Chen, Y., Gershlick, D.C., Park, S.Y., and Bonifacino, J.S. (2017). Segregation in the Golgi complex precedes export of endolysosomal proteins in distinct transport carriers. *J. Cell Biol.* 216, 4141–4151.
- Davis-Kaplan, S.R., Ward, D.M., Shiflett, S.L., and Kaplan, J. (2004). Genome-wide Analysis of Iron-dependent Growth Reveals a Novel Yeast Gene Required for Vacuolar Acidification. *J. Biol. Chem.* 279, 4322–4329.
- Dong, R., Saheki, Y., Swarup, S., Lucast, L., Harper, J.W., and De Camilli, P. (2016). Endosome-ER Contacts Control Actin Nucleation and Retromer Function through VAP-Dependent Regulation of PI4P. *Cell* 166, 408–423.
- Eden, E.R., White, I.J., Tsapara, A., and Futter, C.E. (2010). Membrane contacts between endosomes and ER provide sites for PTP1B-epidermal growth factor receptor interaction. *Nat. Cell Biol.* 12, 267–272.
- Eisenberg-Bord, M., Shai, N., Schuldiner, M., and Bohnert, M. (2016). A Tether Is a Tether Is a Tether: Tethering at Membrane Contact Sites. *Dev. Cell* 39, 395–409.
- Fischer, M.A., Temmerman, K., Ercan, E., Nickel, W., and Seedorf, M. (2009). Binding of plasma membrane lipids recruits the yeast integral membrane protein Ist2 to the cortical ER. *Traffic Cph. Den.* 10, 1084–1097.
- Friedman, J.R., Dibenedetto, J.R., West, M., Rowland, A.A., and Voeltz, G.K. (2013). Endoplasmic reticulum-endosome contact increases as endosomes traffic and mature. *Mol. Biol. Cell* 24, 1030–1040.
- Guyenet, S.J., Furrer, S.A., Damian, V.M., Baughan, T.D., La Spada, A.R., and Garden, G.A. (2010). A Simple Composite Phenotype Scoring System for Evaluating Mouse Models of Cerebellar Ataxia. *J. Vis. Exp. JoVE.*
- Gyobu, S., Ishihara, K., Suzuki, J., Segawa, K., and Nagata, S. (2017). Characterization of the scrambling domain of the TMEM16 family. *Proc. Natl. Acad. Sci. U. S. A.* 114, 6274–6279.

- Hatzipetros, T., Kidd, J.D., Moreno, A.J., Thompson, K., Gill, A., and Vieira, F.G. (2015). A Quick Phenotypic Neurological Scoring System for Evaluating Disease Progression in the SOD1-G93A Mouse Model of ALS. *J. Vis. Exp. JoVE*.
- Jha, A., Chung, W.Y., Vachel, L., Maleth, J., Lake, S., Zhang, G., Ahuja, M., and Muallem, S. (2019). Anoctamin 8 tethers endoplasmic reticulum and plasma membrane for assembly of Ca²⁺ signaling complexes at the ER/PM compartment. *EMBO J.* 38, e101452.
- Kilpatrick, B.S., Eden, E.R., Hockey, L.N., Yates, E., Futter, C.E., and Patel, S. (2017). An Endosomal NAADP-Sensitive Two-Pore Ca²⁺ Channel Regulates ER-Endosome Membrane Contact Sites to Control Growth Factor Signaling. *Cell Rep.* 18, 1636–1645.
- Kim, A.Y., Tang, Z., Liu, Q., Patel, K.N., Maag, D., Geng, Y., and Dong, X. (2008). Pirt, a phosphoinositide-binding protein, functions as a regulatory subunit of TRPV1. *Cell* 133, 475–485.
- Lalonde, R., and Strazielle, C. (2011). Brain regions and genes affecting limb-clasping responses. *Brain Res. Rev.* 67, 252–259.
- Lim, C.-Y., Davis, O.B., Shin, H.R., Zhang, J., Berdan, C.A., Jiang, X., Counihan, J.L., Ory, D.S., Nomura, D.K., and Zoncu, R. (2019). ER-lysosome contacts enable cholesterol sensing by mTORC1 and drive aberrant growth signaling in Niemann-Pick type C. *Nat. Cell Biol.* 21, 1206–1218.
- Lu, Y., Dong, S., Hao, B., Li, C., Zhu, K., Guo, W., Wang, Q., Cheung, K.-H., Wong, C.W., Wu, W.-T., et al. (2014). Vacuolin-1 potently and reversibly inhibits autophagosome-lysosome fusion by activating RAB5A. *Autophagy* 10, 1895–1905.
- Malvezzi, M., Chalal, M., Janjusevic, R., Picollo, A., Terashima, H., Menon, A.K., and Accardi, A. (2013). Ca²⁺-dependent phospholipid scrambling by a reconstituted TMEM16 ion channel. *Nat. Commun.* 4, 2367.
- Manford, A.G., Stefan, C.J., Yuan, H.L., Macgurn, J.A., and Emr, S.D. (2012). ER-to-plasma membrane tethering proteins regulate cell signaling and ER morphology. *Dev. Cell* 23, 1129–1140.
- McKnight, N.C., Zhong, Y., Wold, M.S., Gong, S., Phillips, G.R., Dou, Z., Zhao, Y., Heintz, N., Zong, W.-X., and Yue, Z. (2014). Beclin 1 Is Required for Neuron Viability and Regulates Endosome Pathways via the UVRAG-VPS34 Complex. *PLOS Genet.* 10, e1004626.
- Neefjes, J., and van der Kant, R. (2014). Stuck in traffic: an emerging theme in diseases of the nervous system. *Trends Neurosci.* 37, 66–76.
- Pelz, T., Drose, D.R., Fleck, D., Henkel, B., Ackels, T., Spehr, M., and Neuhaus, E.M. (2018). An ancestral TMEM16 homolog from *Dictyostelium discoideum* forms a scramblase. *PLOS ONE* 13, e0191219.
- Pi, H., Li, M., Tian, L., Yang, Z., Yu, Z., and Zhou, Z. (2017). Enhancing lysosomal biogenesis and autophagic flux by activating the transcription factor EB protects against cadmium-induced neurotoxicity. *Sci. Rep.* 7, 43466.

Raiborg, C., Wenzel, E.M., and Stenmark, H. (2015a). ER–endosome contact sites: molecular compositions and functions. *EMBO J.* *34*, 1848–1858.

Raiborg, C., Wenzel, E.M., Pedersen, N.M., Olsvik, H., Schink, K.O., Schultz, S.W., Vietri, M., Nisi, V., Bucci, C., Brech, A., et al. (2015b). Repeated ER-endosome contacts promote endosome translocation and neurite outgrowth. *Nature* *520*, 234–238.

Rocha, N., Kuijl, C., van der Kant, R., Janssen, L., Houben, D., Janssen, H., Zwart, W., and Neefjes, J. (2009). Cholesterol sensor ORP1L contacts the ER protein VAP to control Rab7-RILP-p150 Glued and late endosome positioning. *J. Cell Biol.* *185*, 1209–1225.

Sandvig, K., and van Deurs, B. (2002). Membrane traffic exploited by protein toxins. *Annu. Rev. Cell Dev. Biol.* *18*, 1–24.

Scorrano, L., Matteis, M.A.D., Emr, S., Giordano, F., Hajnóczky, G., Kornmann, B., Lackner, L.L., Levine, T.P., Pellegrini, L., Reinisch, K., et al. (2019). Coming together to define membrane contact sites. *Nat. Commun.* *10*, 1–11.

Seaman, M.N.J., Michael McCaffery, J., and Emr, S.D. (1998). A Membrane Coat Complex Essential for Endosome-to-Golgi Retrograde Transport in Yeast. *J. Cell Biol.* *142*, 665–681.

Sleigh, J.N., Burgess, R.W., Gillingwater, T.H., and Cader, M.Z. (2014). Morphological analysis of neuromuscular junction development and degeneration in rodent lumbrical muscles. *J. Neurosci. Methods* *227*, 159–165.

Spinosa, M.R., Progida, C., Luca, A.D., Colucci, A.M.R., Alifano, P., and Bucci, C. (2008). Functional Characterization of Rab7 Mutant Proteins Associated with Charcot-Marie-Tooth Type 2B Disease. *J. Neurosci.* *28*, 1640–1648.

Toulmay, A., and Prinz, W.A. (2011). Lipid transfer and signaling at organelle contact sites: the tip of the iceberg. *Curr. Opin. Cell Biol.* *23*, 458–463.

Wang, C., Telpoukhovskaia, M.A., Bahr, B.A., Chen, X., and Gan, L. (2018). Endo-lysosomal dysfunction: a converging mechanism in neurodegenerative diseases. *Curr. Opin. Neurobiol.* *48*, 52–58.

Yang, H., Kim, A., David, T., Palmer, D., Jin, T., Tien, J., Huang, F., Cheng, T., Coughlin, S.R., Jan, Y.N., et al. (2012). TMEM16F Forms a Ca²⁺-Activated Cation Channel Required for Lipid Scrambling in Platelets during Blood Coagulation. *Cell* *151*, 111–122.

Yu, K., Whitlock, J.M., Lee, K., Ortlund, E.A., Cui, Y.Y., and Hartzell, H.C. (2015). Identification of a lipid scrambling domain in ANO6/TMEM16F. *ELife* *4*, e06901.

REVIEWERS' COMMENTS

Reviewer #2 (Remarks to the Author):

The authors have satisfactorily responded to the comments. Thanks!

Reviewer #3 (Remarks to the Author):

The authors answered my questions and I support publication.

Response to reviewers' comments

We would like to thank the reviewers for their time and efforts dedicated to our manuscript. We are happy they find our revised paper suitable for *Nature Communication*.

Point by point response to Reviewer Comments:

Reviewer #2 (Remarks to the Author):

The authors have satisfactorily responded to the comments. Thanks!

Thank you!

Reviewer #3 (Remarks to the Author):

The authors answered my questions and I support publication.

Thank you!